JCB Journal of Cell Biology

# Productive HIV-1 infection of tissue macrophages by fusion with infected CD4+ T cells

Rémi Mascarau[1,2], Marie Woottum[3]*, Léa Fromont[1]*, Rémi Gence[4], Vincent Cantaloube-Ferrieu[5], Zoï Vahlas[1,2], Kevin Lévêque[1], Florent Bertrand[1], Thomas Beunon[1], Arnaud Métais[1], Hicham El Costa[5], Nabila Jabrane-Ferrat[5], Yohan Gallois[6], Nicolas Guibert[7], Jean-Luc Davignon[8], Gilles Favre[4], Isabelle Maridonneau-Parini[1,2], Renaud Poincloux[1,2], Bernard Lagane[5], Serge Bénichou[3], Brigitte Raynaud-Messina[1,2]**, and Christel Vérollet[1,2]**

**Macrophages are essential for HIV-1 pathogenesis and represent major viral reservoirs. Therefore, it is critical to understand macrophage infection, especially in tissue macrophages, which are widely infected in vivo, but poorly permissive to cell-free infection. Although cell-to-cell transmission of HIV-1 is a determinant mode of macrophage infection in vivo, how HIV-1 transfers toward macrophages remains elusive. Here, we demonstrate that fusion of infected CD4+ T lymphocytes with human macrophages leads to their efficient and productive infection. Importantly, several tissue macrophage populations undergo this heterotypic cell fusion, including synovial, placental, lung alveolar, and tonsil macrophages. We also find that this mode of infection is modulated by the macrophage polarization state. This fusion process engages a specific short-lived adhesion structure and is controlled by the CD81 tetraspanin, which activates RhoA/ROCK-dependent actomyosin contractility in macrophages. Our study provides important insights into the mechanisms underlying infection of tissue-resident macrophages, and establishment of persistent cellular reservoirs in patients.**

## Introduction

Although CD4+ T lymphocytes are considered as the main targets of HIV infection, macrophages also play a significant role in HIV-1 infection and contribute to viral persistence and pathogenesis (Hendricks et al., 2021; Sattentau and Stevenson, 2016). They are one of the primary targets for HIV-1 in vivo, as they express the viral CD4 receptor and the chemokine coreceptors CCR5 and CXCR4, although at a lower level than CD4+ T cells. Whereas most of HIV-infected CD4+ T cells die within a few days of infection, in vitro studies suggest that HIV-1-infected macrophages developed mechanisms to limit cell death, resulting in viral replication for extended periods of time. Animal models of infection such as Simian Immunodeficiency Virus (SIV)-infected macaques and HIV-1-infected humanized mice further support in vivo infection of macrophages (Arainga et al., 2017; Avalos et al., 2017; DiNapoli et al., 2017; Honeycutt et al., 2017). In addition, a delayed viral rebound was observed in the context of combination of antiretroviral therapy (cART) in a model of humanized mice reconstituted with only myeloid cells, consistent with the establishment of persistent infection in tissue macrophages (Honeycutt et al., 2017). Infected myeloid cells, especially macrophages, are found as multinucleated giant cells (MGCs; Compton and Schwartz, 2017; Orenstein, 2000; Verollet et al., 2010), in diverse tissues of HIV-infected patients (Rodrigues et al., 2017; Sattentau and Stevenson, 2016), including lymph nodes, spleen, lungs, genital, digestive tracts, and the brain. As CD4+ T lymphocytes (Murooka et al., 2012; Stolp et al., 2009), macrophages are also involved in virus dissemination in numerous host tissues (Verollet et al., 2015). Finally, they participate in chronic activation of the immune system, in particular because they resist to CD8+ T cell- and natural killer-mediated killing (Clayton et al., 2018; Clayton et al., 2021). Macrophages thus promote the establishment of persistent viral reservoirs in tissues (Ganor et al., 2019; Hendricks et al., 2021), potentially being a source of rebound viremia following cART cessation (Andrade et al., 2020).

Macrophages exhibit great heterogeneity (Bleriot et al., 2020; Murray, 2017). In response to the local cytokine environment,

[1]Institut de Pharmacologie et Biologie Structurale (IPBS), Université de Toulouse, Centre National de la Recherche Scientifique, Université Toulouse III - Paul Sabatier (UPS), Toulouse, France; [2]International Research Project " MAC-TB/HIV ", Toulouse, France; [3]Institut Cochin, Inserm U1016, Centre National de la Recherche Scientifique UMR8104, Université de Paris, Paris, France; [4]Centre de Recherches en Cancérologie de Toulouse, Inserm UMR1037 et Institut Universitaire du Cancer de Toulouse - Oncopôle, Toulouse, France; [5]Institut Toulousain des Maladies Infectieuses et Inflammatoires, Université Toulouse, Centre National de la Recherche Scientifique, Inserm, Toulouse, France; [6]ENT, Otoneurology and Pediatric ENT Department, University Hospital of Toulouse, Toulouse, France; [7]Thoracic Endoscopy Unit, Pulmonology Department, Larrey University Hospital, Toulouse, France; [8]Parc Scientifique de Luminy, UMRS1097, Marseille, France.

*M. Woottum. and L. Fromont contributed equally to this paper; **B. Raynaud-Messina and C. Vérollet contributed equally to this paper. Correspondence to Brigitte Raynaud-Messina: raynaud@ipbs.fr; Christel Vérollet: verollet@ipbs.fr.

macrophages can be derived toward pro-inflammatory (M1) or anti-inflammatory (M2) phenotypes. In vivo, it is likely that a polarization switch occurs from a dominant M1 program during the acute phase of infection to the M2 programs through later stages (Cassol et al., 2010; Lugo-Villarino et al., 2011). In vitro, both M1 and M2 are less susceptible to HIV infection with cell-free particles in comparison with unpolarized conditions, albeit using different mechanisms (Cassol et al., 2009; Cobos Jimenez et al., 2012). This is consistent with the well-known refractory state toward HIV-1 infection of tissue macrophages (Hendricks et al., 2021; Johnson and Chakraborty, 2016; Schiff et al., 2021). The other mode of infection by virus cell-to-cell transfer is more efficient than infection with cell-free viruses and appears as the major determinant for virus dissemination in vivo (Dupont and Sattentau, 2020; Murooka et al., 2012; Sewald et al., 2012). It is particularly relevant regarding macrophages, which are modestly infected by cell-free viruses (Cassol et al., 2010; Han et al., 2022; Hendricks et al., 2021). Importantly, this mode of virus transmission enables the virus to escape cART and elimination by the immune system, including neutralizing antibodies (Schiffner et al., 2013). However, little is known about the mechanisms that control virus dissemination by cell-to-cell transfer for macrophage infection. Several mechanisms have been evidenced such as the establishment of the virological (Jolly et al., 2004; Jolly et al., 2007a; Jolly et al., 2007b; Jolly and Sattentau, 2007) and infectious synapses (Izquierdo-Useros et al., 2012; Menager and Littman, 2016; Nikolic et al., 2011) that allow cis- and trans-infection, respectively, or the formation of tunneling nanotubes (Dupont et al., 2020; Souriant et al., 2019; Sowinski et al., 2008), but the molecular mechanisms, the efficiency, and the relevance in vivo of these processes are not well understood. Despite the great diversity of mechanisms allowing intercellular transfer of the virus, they converge with the involvement of the actin cytoskeleton and the formation of adhesion-like structures (Compton and Schwartz, 2017; Dufrancais et al., 2021; Lehmann et al., 2011).

Since CD4+ T cells are the primary type of cells infected throughout infection and HIV-1 isolates infecting macrophages are rare, viral transfer from infected CD4+ T cells to macrophages is the most likely mode of transmission, which can account for the systemic distribution of infected macrophages in vivo (Hendricks et al., 2021; Joseph and Swanstrom, 2018). In vitro, two mechanisms have been described for macrophage infection by virus cell-to-cell transfer from infected T lymphocytes: phagocytosis of the infected T cells by macrophages (Baxter et al., 2014) and heterotypic cell fusion of infected T cells with macrophages (Bracq et al., 2017; Han et al., 2022; Xie et al., 2019). The latter has been well characterized at the virological point of view (Bracq et al., 2017; Han et al., 2022; Xie et al., 2019). HIV-1 Infection of macrophages by heterotypic cell fusion requires interaction between the surface viral envelope glycoprotein gp120 expressed on infected T cells and CD4/CCR5 on the macrophage targets, as it was totally abrogated by anti-CD4 and anti-gp120 antibodies, T20, or Maraviroc (Bracq et al., 2017). This process of cell fusion favors a massive transfer of the virus to macrophages that is no longer controlled by the restriction factor SAMHD1 (Xie et al., 2019). Importantly, this viral

transmission mode by fusion is also able to overcome the entry block of isolates initially defined as non-macrophage-tropic (Han et al., 2022). Infection of macrophages by cell-to-cell transfer as well as by cell-free viruses renders them capable of subsequent homotypic cell fusion with neighboring uninfected macrophages, resulting in the formation of infected MGCs (Bracq et al., 2018; Bracq et al., 2017).

In the present study, we investigated the mechanisms involved in macrophage infection by virus transfer from infected CD4+ T cells. We show that the phagocytosis-mediated mechanism happens primarily when HIV-1-infected T cells are apoptotic. A more productive viral infection of macrophages is observed through heterotypic cell fusion with viable infected T cells. We then reveal the relevance of this process in macrophages from different tissues and its modulation by the cytokine environment. Finally, we show that the CD81 tetraspanin is a negative regulator of cell fusion between HIV-1 infected CD4+ T cells and macrophages by acting through the RhoA-ROCK/Myosin II axis.

## Results

### Heterotypic cell fusion of HIV-1-infected T lymphocytes with macrophages is the most efficient mode of cell-to-cell viral transfer

We first evaluated the efficacy of the two already described modes of macrophage infection by virus cell-to-cell transfer from T lymphocytes, which are phagocytosis and cell fusion (Baxter et al., 2014; Bracq et al., 2017). For this purpose, we established experimental conditions that promote either of the major modes of virus transmission by affecting the viability of infected virus-donor Jurkat T cells. Jurkat cells were first infected with the NLAD8 HIV-1 strain during 2 or 7 d, and then co-cultured for 24 h with primary human monocytes-derived macrophages (MDMs) at a 1:1 ratio (Fig. 1 A). After 2 d infection, Jurkat cells were usually around 20–30% infected and almost all alive (≥95%; Fig. S1 A), this low mortality being mostly due to a bystander effect (Fig. S1B and Matrajt et al., 2014). In contrast, Jurkat cells infected for 7 d were more than 90% apoptotic, with an infection rate quite similar to that of Jurkat cells infected for 2 d (Fig. S1 A). After co-culture, immunofluorescence (IF) analysis of MDMs and the use of different markers allowed the identification of MDM infection in three distinct modes (Fig. S1, C–F). The first one, i.e., cell-to-cell virus transfer by phagocytosis, showed a moderate HIV-p24 signal in infected MDMs colocalizing with low pH compartments (as evidenced by Lysotracker or pHrodo staining) and T cell markers (Baxter et al., 2014; Fig. S1, C–E). The second mode of MDM infection by cell fusion was characterized by the formation of multinucleated cells (Fig. S1 F) with at least one T cell nucleus and a strong and diffuse cytoplasmic HIV-p24 signal, without specific accumulation in lysosomal compartments (Fig. S1, C–E). Cell fusion was confirmed by pre-labeling Jurkat cells and MDMs with different cytoplasmic CellTrackers (Fig. 1 B and Video 1) and the presence of a T cell marker (CD3) in most HIV-p24+ cells by flow cytometry analysis (Fig. S2 A). Finally, we defined a synapse-like mode of virus transfer to MDMs corresponding to

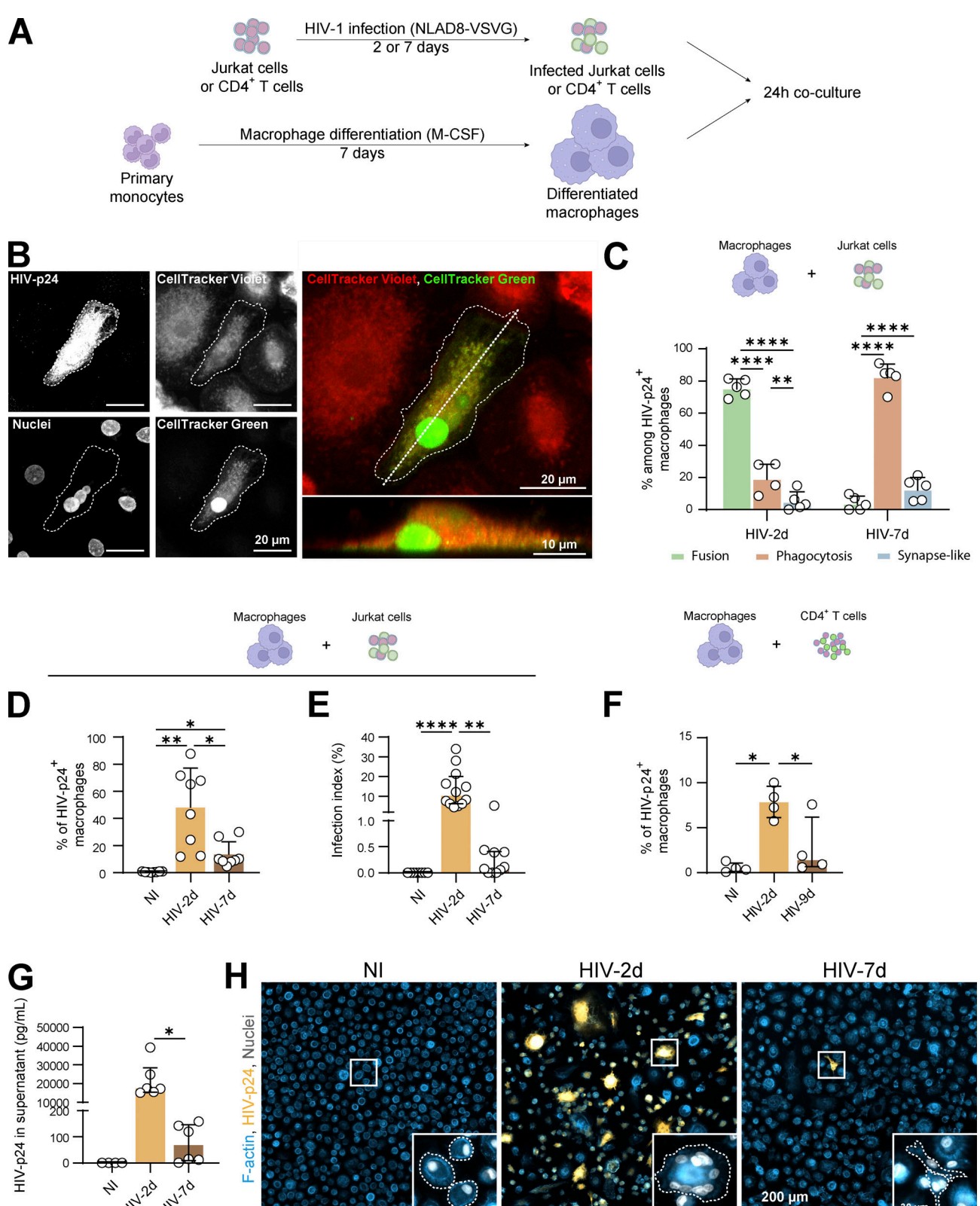

Figure 1. **Heterotypic fusion between infected T cells and macrophages is the most productive mode of HIV-1 cell-to-cell transfer toward macrophages. (A)** Experimental design. Human monocytes isolated from blood were differentiated into macrophages for 7 d (MDMs), co-cultured during 24 h with CD4[+] T cells (Jurkat cell line or autologous primary CD4[+] T cells) infected with HIV-1 (NLAD8-VSVG) (for 2 or 7 d), CD4[+] T cells were eliminated and MDM infection was analyzed. **(B)** Representative confocal image of a MDM pre-loaded with CellTracker Violet (red) infected by fusion with 2 d-infected Jurkat cells pre-loaded with CellTracker (green), HIV-p24 and nuclei (DRAQ5) are shown in gray. Merge: z projection along the dotted line is shown in lower panel. Scale bars, 20 and 10 µm. See also Video 1. **(C)** Quantification of the three mechanisms (fusion, phagocytosis, and synapse-like) of HIV-1 transfer from infected CD4[+]

T cells to MDMs as characterized in Fig. S1, C and D, among infected MDMs after a 24 h-co-culture with Jurkat cells infected for 2 d (HIV-2d) or 7 d (HIV-7d; n = 4 or 5 donors, mean ± SD). **(D and E)** Analysis of MDM infection after a 24 h-co-culture with uninfected Jurkat cells (NI) or Jurkat cells infected for 2 d (HIV-2d) or 7 d (HIV-7d) by flow cytometry (D, % of HIV-1-p24 + MDMs, n = 8 donors, mean ± SD) and fluorescence microscopy (E, MDM infection index, n = 11 donors, median ± interquartile range). **(F)** Analysis of MDM infection after a 24 h co-culture with uninfected autologous T cells (NI) or autologous primary CD4⁺ T cells infected for 2 d (HIV-2d) or 9 d (HIV-9d) by flow cytometry (n = 4 donors, median ± interquartile range). **(G and H)** Analysis of long-term MDM infection. After co-culture as in A, Jurkat cells were washed, and MDMs were kept in culture for additional 5 d. **(G)** Quantification of HIV-p24 concentration in the supernatant of MDMs (n = 6 donors, median ± interquartile range). **(H)** Representative microscopy images of MDM infection. F-actin (phalloidin, blue), HIV-p24 (yellow), nuclei (DAPI, gray). Inserts show magnifications of the white squares. Scale bars, 200 and 30 μm. Statistical analyses: Multiple comparison test (C) two-way Anova and Tukey, (D) one-way Anova and Tukey, (E–G) Kruskal-Wallis and Dunn's, and (F) Friedman and Dunn's. * P ≤ 0.05; ** P ≤ 0.01; **** P ≤ 0.0001.

the presence of infected mononucleated MDMs with a faint and dotted HIV-p24 signal outside the acidic compartments (Fig. S1, C, D, and F). As shown in Fig. 1 C, when co-cultured with 2-d-infected living T cells (HIV-2d), MDMs were mainly infected by cell fusion, whereas phagocytosis became the main viral transmission route when MDMs were infected by co-culture with 7-d-apoptotic-infected T cells (HIV-7d). In both cases, the synapse-like mechanism of infection remained minor (<15%).

Importantly, when MDMs were co-cultured with 2-d-infected living Jurkat cells and thus mainly infected through cell fusion, around 50% of MDMs were positive for HIV-p24 by flow cytometry analysis, whereas p24 staining was only observed in up to 10% of MDMs co-cultured with 7-d-apoptotic infected Jurkat cells (Fig. 1 D and Fig. S2 B). A similar trend of MDM infection was obtained by IF (Fig. 1 E and Fig. S2 C). Additional experiments showed that infection of MDMs was more effective when co-cultured with primary autologous CD4⁺ T cells infected for 2 d compared with infected counterparts for 9 d (Fig. 1 F and Fig. S2 D). The increase in the efficiency of MDM infection observed when MDMs were co-cultured with living T cells compared with apoptotic Jurkat T cells was even exacerbated at day 5, as evidenced by the marked increase of virus release through quantification of the HIV-p24 in cell-culture supernatants (Fig. 1 G), as well as the higher number of infected cells (Fig. 1 H). The viruses produced by the fusion mechanism were infectious as they can infect the TZM-bl reporter cell line, and their infectivity was comparable to that of the few viruses produced by phagocytosis (12.2 ± 3.1% of p24-positive TZM-bl vs. 9.3 ± 4.5%, respectively; n = 3).

Altogether these data show that the viability of infected virus-donor CD4⁺ T cell dictates the mode of virus transmission toward macrophages, and that the heterotypic cell fusion mechanism, largely favored with living infected T lymphocytes, is by far the most effective mode of infection.

### Human tissue macrophages are permissive to infection by HIV-1-induced cell fusion

We then investigated whether this viral transmission mode by heterotypic cell fusion occurs in macrophages from human tissues in which macrophages are susceptible to encounter infected CD4⁺ T lymphocytes. First, ex vivo explants of non-inflammatory synovial membrane that contain moderate level of lymphocytes and macrophages (Raynaud-Messina et al., 2018) were exposed to 2-d-infected Jurkat T cells. After 24 h, we observed that some CD68⁺ macrophages were positive for HIV-p24, multinucleated and positive for the CD3 T cell marker, demonstrating that in situ infection of macrophages by fusion

with HIV-1-infected Jurkat cells was possible (Fig. 2 A and Fig. S3 A). We then completed our study with other models of purified human tissue macrophages (Fig. 2, B and C; Fig. S3 B; and Fig. S4). We first purified alveolar macrophages that constitute the largest proportion of cells within broncho-alveolar lavages (Salahuddin et al., 2019). Although very fragile and uneasy to handle, these macrophages were efficiently infected via cell fusion with infected Jurkat T cells, as revealed by the presence of multinucleated macrophages strongly stained for the viral protein HIV-p24 and containing at least one T cell nucleus (Fig. 2 B, Video 2, and Fig. S3 B, upper panels). We verified that this infection mode of alveolar macrophages was sensitive to pre-incubation with anti-CD4 antibodies (Fig. S3 B, lower panels). Then, we set up a protocol to purify the rare myeloid cells among the mononuclear cell population isolated from healthy tonsils (Smith et al., 2020). These cells were also infected via heterotypic cell fusion after 24 h-co-culture with infected Jurkat T cells. Indeed, all the HIV-p24-positive macrophages observed were multinucleated and contained one nucleus from T cells (Fig. 2 C and Video 3). Finally, placental macrophages, known to be involved in transmission of many pathogens including HIV-1 (El Costa et al., 2016a; Espino et al., 2021), express HIV-1 receptors and co-receptors but are weakly permissive to cell-free infection compared to MDMs (Johnson and Chakraborty, 2016). Here, we show that CD14⁺ macrophages purified from first-trimester placenta were infected through heterotypic fusion with infected Jurkat T cells (Fig. S4, A and B). This infection mode was more efficient than cell-free virus infection and dependent on CD4 expression on target macrophages as we observed a 75% inhibition in the infection index when macrophages were pre-treated with anti-CD4 antibodies (n > 270 nuclei/condition; Fig. S4, A and B).

Altogether, these data show that human tissue macrophages, independently of their tissue localization, are permissive to infection by heterotypic cell fusion with viable infected Jurkat T cells.

### Anti-inflammatory activation exacerbates HIV-1 infection of macrophages through heterotypic fusion with infected T cells

Given the wide range of macrophage activation states depending on tissue distribution and HIV-1 disease progression (Cassol et al., 2010), and considering previous data showing that macrophage polarization is a critical factor for infection efficiency by cell-free virus (Cassol et al., 2010; Cassol et al., 2009; Cobos Jimenez et al., 2012; Hendricks et al., 2021; Lugo-Villarino et al., 2011; Trus et al., 2020), we evaluated whether different activation programs of MDMs generated in vitro could modulate their

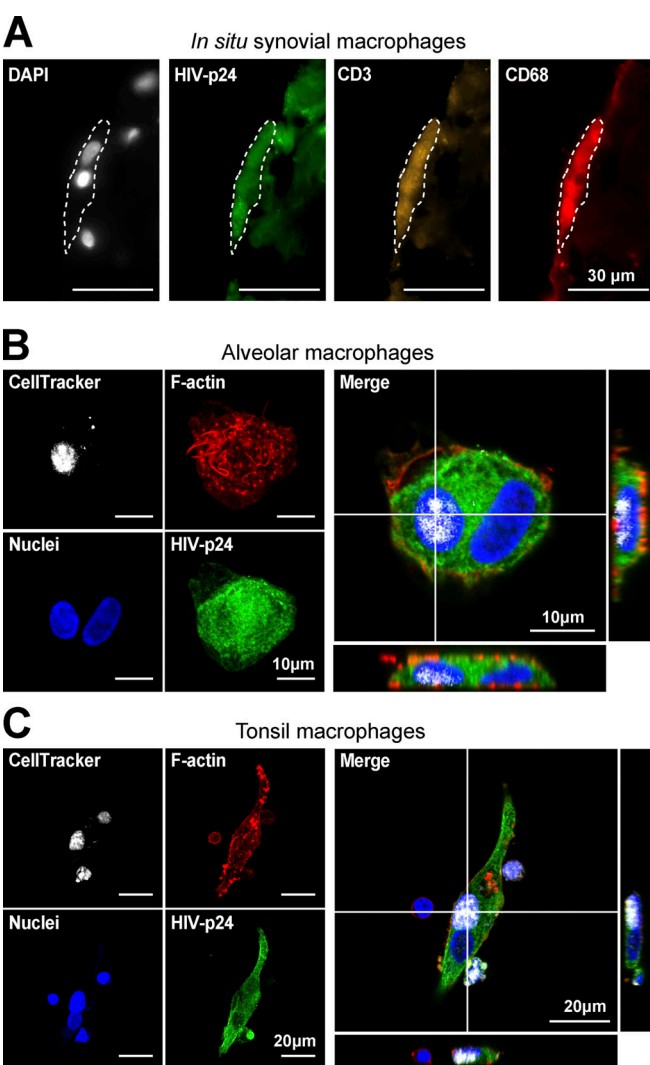

**Figure 2. HIV-1-induced heterotypic cell fusion with infected Jurkat T cells is relevant in several human tissue macrophages. (A)** In situ tissue macrophage infection: Non-inflammatory human synovial explants were incubated with 2 d-infected Jurkat cells during 24 h and processed for immunohistological analyses. Nuclei (DAPI, gray), HIV-p24 (green), CD3 (CD4+ T cell marker, orange), and CD68 (macrophage marker, red). Scale bar, 30 µm. **(B and C)** Ex vivo infection of purified tissue macrophages: Macrophages isolated from adult broncho-alveolar lavages (B, see also Video 2) or children hyperplasic tonsils (C, see also Video 3) were co-cultured for 24 h with 2-d-infected Jurkat cells pre-loaded with CellTracker (gray) and analyzed by confocal microscopy. Representative confocal microscopy images with HIV-p24 (green), F-actin (phalloidin, red), and nuclei (DAPI, blue). Scale bars, 10 µm (B) and 20 µm (C).

ability to fuse with infected T cells. Before the co-culture with 2-d-infected Jurkat cells, we first polarized monocytes during 7 d, with GM-CSF or M-CSF leading to M1-and M2-like MDMs, respectively (Trus et al., 2020). Compared to GM-CSF, M-CSF-driven conditions strongly favored HIV-1 infection of MDMs (Fig. S5 A). IF analysis also revealed that heterotypic fusion was the main infection mechanism in M2-like MDMs (M(M-CSF)), while the synapse-like mode of infection appeared to be favored in pro-inflammatory conditions at the expense of the cell fusion mechanism (Fig. S5 B). In the two conditions, infection through

phagocytosis remained minor. To go further, we generated extreme M1 (M(IFNγ/LPS)) and M2 (M(IL-4) and M(IL-10)) phenotypes (Fig. 3 A). After co-culture with 2-d-infected Jurkat cells, a significant increase of infectivity in M(IL-4)) and M(IL-10) was observed compared to M(IFN-γ/LPS; Fig. 3, B and C). Importantly, we obtained similar results using human MDMs co-cultured with infected autologous primary CD4+ T cells (Fig. 3 F). Microscopy analysis confirmed the higher infection rate of anti-inflammatory MDMs compared with their pro-inflammatory counterparts when co-cultured with infected Jurkat cells and revealed a strong correlation between the infection index and the fusion index (Fig. 3, D and E; and Fig. S5 C). A similar pattern of infection was obtained when MDM infection was mediated by HIV+ autologous primary CD4+ T cells (Fig. S5 D). The decrease of infection efficiency in pro-inflammatory conditions was associated with a significant reduction of the surface expression of CD4 in M(IFN-γ/LPS) compared to other polarization profiles, while the level of the CCR5 co-receptor was not affected (Fig. S5 E; Cassol et al., 2009). Upon 5 d after the co-culture, the ability of M(IFN-γ/LPS) MDMs to be infected and to sustain HIV-1 production was around 10-fold lower than that of M(M-CSF), M(IL-4)) or M(IL-10) MDMs as measured by IF and p24 release in the culture supernatants (Fig. 3, G and H). The viruses produced in each condition were equally infectious on TZM-bl reporter cells.

These results show that the activation profile of macrophages influences their mode of infection, anti-inflammatory environments promoting infection of macrophages by heterotypic fusion with infected CD4+ T cells.

### Actomyosin contractility controls heterotypic fusion of macrophages with HIV-1-infected T cells

Since cell fusion is specific to contacts between infected T cells and macrophages and not triggered in the context of the T cell–T cell virological synapse (Sattentau, 2008), we determined the molecular mechanisms involved in this mode of infection focusing on the target macrophages. We therefore first performed live imaging of MDMs after contact with Jurkat T cells infected with a GFP-tagged virus. This contact, which typically lasted <20 min, resulted in a strong and homogeneous GFP signal inside the heterokaryon attesting for the presence of the virus and the fusion of the two cells (Fig. 4 A and Video 4). The short interaction observed before fusion was in marked contrast with the synapse-like mode of infection showing a cell contact that could last for more than 10 h and resulted in a punctuated GFP signal in infected mononucleated MDMs (Fig. S6 A and Video 5). To further characterize the fusogenic contacts, we used a combination of scanning electron and confocal microscopy. In contrast to MDMs in contact with uninfected Jurkat cells, MDMs co-cultured with infected Jurkat T cells became polarized with a deformation of the cell body where cell interaction occurred (Fig. 4 B). Moreover, confocal analysis revealed an increased recruitment of β2 integrin/CD18 and F-actin at cell contacts (Fig. 4, C and D), enhancing evidence for specific interactions between MDMs and infected T cells compared to uninfected T cells. Blocking CD18 with specific antibodies significantly reduced the infection of MDMs by fusion with infected T cells

**Figure 3. Macrophage polarization drives their ability to fuse with HIV-1 infected T lymphocytes. (A)** Experimental design for MDM polarization. Human monocytes were differentiated in presence of GM-CSF (M(GM-CSF)) or M-CSF (M(M-CSF)) for 7 d and incubated for additional 2 d with either lipopolysaccharide and interferon-γ (M(LPS+IFNγ)), interleukin-4 (M(IL-4)), or interleukin-10 (M(IL-10)). **(B–F)** Analysis of HIV-1 infection of polarized MDMs after a 24 h-co-culture with infected Jurkat cells (B–E) or autologous primary CD4[+] T cells (F) by flow cytometry (B, C, and F) or fluorescence microscopy (D–E). **(B)** Representative dot plots for HIV-1 p24 signal in MDMs and gating strategy for selection of infected cells. **(C and F)** Quantification of MDM infection, normalized to the M(M-CSF) condition. C, n = 9 donors, median ± interquartile range and F, n = 5 donors, mean ± SD. **(D–E)** Quantification of MDM infection index (D, n = 7 donors) and MDM fusion index (E, n = 7 donors). Median ± interquartile range. **(G and H)** Analysis of long-term infection of polarized MDMs. After 24 h of co-culture, Jurkat cells were washed, and polarized MDMs were maintained in culture for additional 5 d. **(G)** Representative microscopy images: F-actin (phalloidin, blue), HIV-p24 (yellow) and nuclei (DAPI, gray). Inserts show magnifications of the white squares. Scale bars, 100 and 50 μm. **(H)** Quantification of HIV-p24 concentration in the supernatant of infected MDMs (n = 6 donors). Median ± interquartile range. Statistical analyses: Multiple comparison test (F) one-way Anova and Tukey, (C) Kruskal–Wallis and Dunn's, and (D, E, and H) Friedman and Dunn's. * P ≤ 0.05; ** P ≤ 0.01.

using either Jurkat or primary autologous CD4[+] T cells (Fig. 4, E and F).

We then determined the impact of different actin-disturbing drugs on cell fusion by pre-treating MDMs with Jasplakinolide, Cytochalasin D, and the Arp2/3 inhibitor CK666. All these drugs

were used at non-cytotoxic concentrations but capable of disrupting the formation or organization of podosomes, the main F-actin adhesion structures in macrophages (Fig. S6 B; Labernadie et al., 2010; Wiesner et al., 2014). They also significantly reduced MDM infection by fusion with infected T cells

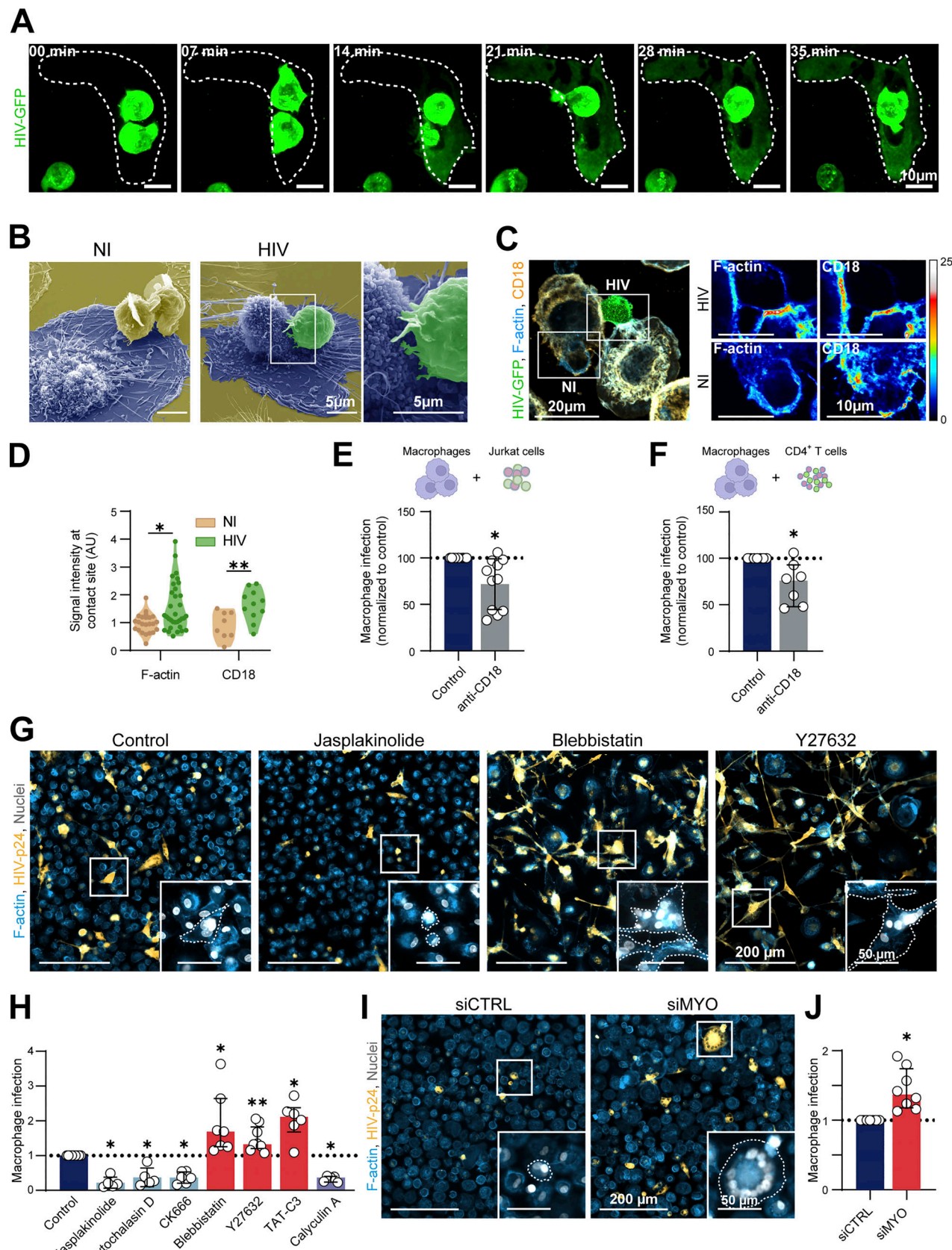

Figure 4. **Characterization of the fusogenic contact and role of actin cytoskeleton in fusion of macrophages with infected Jurkat T cells.** **(A–F)** Characterization of the fusogenic contact between infected Jurkat T cells and MDMs. **(A)** Time-lapse images of MDM infection by fusion: Jurkat cells were infected with a GFP-expressing viral strain (ADA-GFP-VSVG), infected cells were sorted by flow cytometry and co-cultured with MDMs 30 min before

starting live-microscopy (see also Video 4). HIV-GFP in green, dashed white lines delineate the MDM. Scale bars, 10 µm. Representative images of more than 10 cell interactions leading to fusion. **(B)** Representative scanning electron microscopy (SEM) image of MDMs (blue) in co-culture for 30 min with Jurkat cells uninfected (NI, yellow) or infected with HIV-GFP (green). Scale bars, 5 µm. **(C)** Left: Representative confocal microscopy image of MDMs in co-culture for 30 min with a mix of uninfected (NI) and infected (HIV) Jurkat cells. F-actin (phalloidin, blue), HIV-GFP (green), and CD18 (yellow). Right: Twofold magnification of F-actin and CD18 fluorescence intensity (false colors). Scale bars, 20 and 10 µm. **(D)** Quantification of normalized F-actin and CD18 signal intensity at cell contacts between MDMs and Jurkat cells infected (HIV) or not (NI), from images corresponding to Fig. 4 C (for actin, n = 23–33 images and for CD18, n = 8–11 images, from three donors, median ± interquartile range). **(E and F)** Quantification by flow cytometry of infection of MDMs incubated with isotype control (Control) or antibody targeting CD18 (anti-CD18) and co-cultured during 24 h with infected Jurkat cells (C, n = 11 donors) or autologous primary T cells (D, n = 7 donors), normalized to the isotype control condition. Mean ± SD. **(G and H)** MDMs were pre-treated with different inhibitors (Jasplakinolide, Cytochalasin D, CK666, Blebbistatin, Y27632, TAT-C3, or Calyculin A) or DMSO (Control) and co-cultured during 24 h with infected Jurkat cells. **(G)** Representative images of co-cultures treated with Jasplakinolide, Blebbistatin, Y27632, or not (Control). F-actin (phalloidin, blue), HIV-p24 (yellow) and nuclei (DAPI, gray). Scale bar, 200 and 50 µm. **(H)** Quantification by microscopy analysis of MDM infection, normalized to the control condition (n = 6 to 8 donors, median ± interquartile range). **(I and J)** Analysis of infection of MDMs transfected with non-targeting siRNA (siCTRL) or targeting myosin IIA (siMYO) before co-culture with infected Jurkat cells. **(I)** Representative microscopy images of MDM infection. F-actin (phalloidin, blue), HIV-p24 (yellow), and nuclei (DAPI, gray). Scale bars, 200 and 50 µm. **(J)** Quantification of MDM infection by flow cytometry, normalized to the siCTRL condition (n = 8 donors). Mean ± SD. Statistical analyses: (D) Mann-Whitney (F-actin) or unpaired t test (CD18) test, (E, F, and J) Paired t test, (H) Wilcoxon test or Paired t test. * P ≤ 0.05; ** P ≤ 0.01.

(Fig. 4, G and H), attesting that actin cytoskeleton is critical for this heterotypic cell fusion process. Actin cytoskeleton dynamics is strongly regulated by actomyosin complexes. Activation by phosphorylation of the regulatory light chain of myosin (MLC) is mainly mediated by the ROCK kinase which is a major effector of RhoA (Julian and Olson, 2014). To assess the role of myosin activity in the HIV-1-induced heterotypic cell fusion process, MDMs were first pre-treated with either the myosin II inhibitor Blebbistatin or a broad phosphatase inhibitor (Calyculin A) before the co-culture. As expected (Kolega, 2006; Labernadie et al., 2010), upon Blebbistatin and Calyculin A treatments, MDMs showed an altered actomyosin cytoskeleton without any effect on cell viability (as the cell density was comparable with the control condition) or podosome formation (Fig. S6, C and D). Blebbistatin treatment of MDMs induced a two-fold increase in the percentage of multinucleated HIV-p24+ cells, whereas Calyculin A inhibited by threefold the process (Fig. 4, G and H). We then used TAT-C3 and Y27632 that are inhibitors of Rho-GTPases and ROCK, respectively (Fig. 4, G and H; and Fig. S6 C). In the two conditions, MDM infection by cell fusion with infected T cells was significantly increased. Finally, silencing of myosin IIA in MDMs using an efficient siRNA approach (Dupont et al., 2022; Dupont et al., 2020; Troegeler et al., 2017) led to a decrease in MLC phosphorylation (Fig. S7, A and B) along with a significant increase in the number of infected multinucleated MDMs (Fig. 4, I and J), confirming the inhibitory role of myosin IIA in MDM fusion with infected T cells. We verified by IF that siRNA against myosin did not affect cell viability or cell morphology (Fig. S7 C).

Together, these findings demonstrate that actin dynamics and moderate myosin activity inside macrophages act in concert to favor their infection through heterotypic fusion with infected T lymphocytes.

### The macrophage tetraspanin CD81 restrains heterotypic cell fusion through the RhoA/actomyosin pathway

Tetraspanins constitute a large family of transmembrane proteins involved in the organization of specialized membrane microdomains through their ability to interact with numerous partners, including integrins, leukocyte receptors, cytoskeleton, and signaling proteins (Hemler, 2005). Thus, tetraspanins participate in many biological processes, including cell fusion where they exert either inhibitory or activating effects depending on the experimental settings and the identity of the cell partners (Gordon-Alonso et al., 2006; Weng et al., 2009; Whitaker et al., 2019). Upon the 18 tetraspanins expressed by monocytes and macrophages, CD9 and CD81 have been shown to be involved in cell migration and fusion (Takeda et al., 2008; Takeda et al., 2003). We, therefore, investigated the role of these two proteins in macrophage infection through fusion with living infected T cells. While anti-CD9 antibodies had no effect (Fig. S7 D), the anti-CD81 antibody produced a significant increase in the percentage of HIV-p24+ cells by approximately fourfold (Fig. 5 A). In addition, specific CD81 targeting in MDMs using siRNA significantly increased the ability of MDMs to fuse with infected Jurkat T cells (Fig. 5 B and Fig. S7 E), confirming the inhibitory role of macrophage CD81 in cell fusion. Similar results were obtained with MDMs pre-incubated with anti-CD81 antibodies before co-culture with autologous primary CD4+ T cells (Fig. 5 C). As a proof of concept, the inhibitory role of CD81 in macrophage infection via fusion was further confirmed in human macrophages purified from broncho-alveolar lavages (Fig. 5 D and Fig. S7 F). Finally, to determine whether CD81 could exert its inhibitory effect through regulation of the actomyosin cytoskeleton, we assessed the impact of CD81 silencing on the distribution of actomyosin fibers and the molecular pathways involved in myosin activation. CD81 depletion in MDMs induced a slight disruption of cortical myosin labeling and no impact on podosome organization, a similar pattern to the one obtained after Y27632 and TAT-C3 treatments, suggesting the involvement of myosin activation in the inhibitory effect of CD81 (Fig. S7 G). To support this hypothesis, we showed that CD81 silencing in MDMs was correlated with a significant reduction in the level of MLC phosphorylation while the total level of myosin II was not affected (Fig. 5 E). Finally, downregulation of CD81 was associated with a significant reduction in the level of endogenous active RhoA-GTP as measured using a rhotekin-RBD (Rho binding domain) pulldown assay (Fig. 5 F).

Altogether these findings indicate that CD81 is a negative regulator for macrophage infection through RhoA/ROCK activation leading to myosin phosphorylation, which limits heterotypic fusion of macrophages with infected T cells.

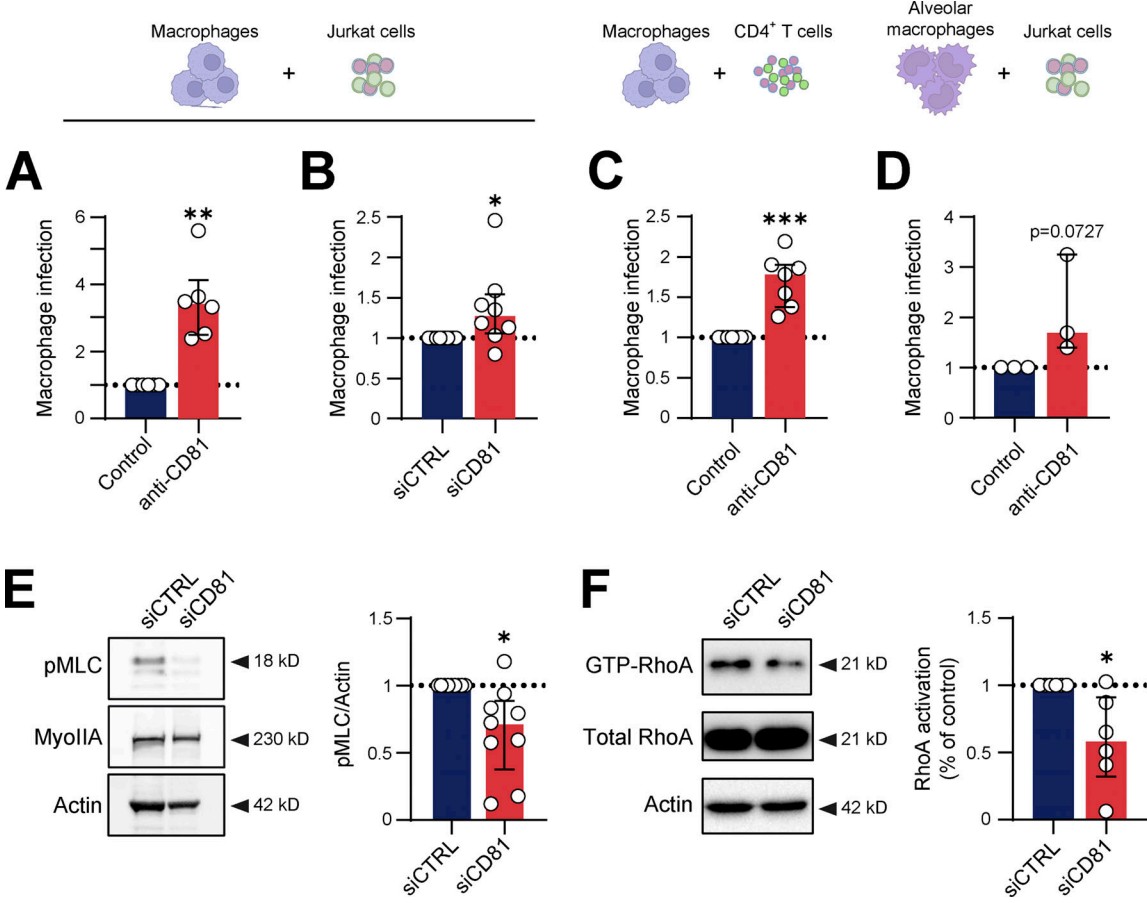

Figure 5. **CD81 negatively regulates heterotypic fusion of macrophages with infected Jurkat T cells through the RhoA/ROCK/actomyosin pathway.** **(A–D)** Role of CD81 in infection of macrophages by fusion with infected CD4[+] T cells. **(A)** Quantification by flow cytometry of infection of MDMs incubated with isotype control (Control) or antibody targeting CD81 (anti-CD81) before co-culture with infected Jurkat cells, normalized to the isotype control condition ($n = 6$ donors, mean ± SD). **(B)** MDMs were transfected with non-targeting siRNA (siCTRL) or targeting CD81 (siCD81) and co-cultured during 24 h with infected Jurkat cells. Quantification by flow cytometry of MDM infection, normalized to the siCTRL condition ($n = 8$ donors, mean ± SD). **(C)** Same experiment as in A except that Jurkat cells were replaced by autologous primary T cells ($n = 7$ donors, mean ± SD). **(D)** Same experiment as in A except that MDMs were replaced by macrophages isolated from broncho-alveolar lavages ($n = 3$ donors, median ± interquartile range). **(E and F)** Involvement of CD81 in the regulation of myosin II activity. **(E)** Quantification of MLC activation. Left: Representative images of Western blot analysis illustrating the phosphorylation of MLC (p-MLC, top) and expression of total myosin IIA (middle), with actin as loading control (bottom). Right: Quantification of p-MLC/actin ratio, normalized to the siCTRL condition ($n = 9$ donors, median ± interquartile range). **(F)** Quantification of RhoA activation: After CD81 silencing, MDMs were lysed and then pull-down of active GTP-Rho was performed using GST-RBD beads. Left: Representative images of Western blot analysis illustrating the expression of active RhoA (GTP-RhoA, top) and total RhoA (middle), both revealed with a RhoA antibody with actin as loading control (bottom). Right: Quantification of GTP-RhoA/total RhoA, normalized to actin; siCTRL condition serves as a reference ($n = 6$ donors, median ± interquartile range). Statistical analyses: (A–C) Paired $t$ test and (D–F) Wilcoxon test. * $P ≤ 0.05$; ** $P ≤ 0.01$.

## Discussion

Macrophages participate in HIV-1 dissemination and establishment of persistent virus reservoirs in numerous host tissues. In this study, we have characterized HIV-1 entry routes of macrophage infection by virus cell-to-cell transfer from infected CD4[+]T lymphocytes and describe how they are regulated. Our findings (Fig. 6) highlight that, among the mechanisms of virus transfer, heterotypic cell fusion between infected T lymphocytes and macrophages is the most productive. Importantly, this mode of infection occurs in several tissue macrophages and is modulated by the macrophage activation state. Finally, the identification of the CD81/RhoA-ROCK/Myosin II signaling cascade as a key player of this process constitutes a major step in understanding the mechanisms of HIV-1 spreading toward macrophages.

In contrast to cell-free HIV-1 infection, cell-to-cell transmission may be the major route of HIV-1 infection and spreading in vivo, particularly in tissues with high HIV-1 target cell density (Bracq et al., 2018; Jolly et al., 2004). First, by comparing the two mechanisms already described for macrophage infection via transfer from infected T cells (Baxter et al., 2014; Bracq et al., 2017), we found that heterotypic cell fusion is much more effective for macrophage infection compared to phagocytosis. The fusion mechanism is strongly favored when infected T cells are alive while macrophage infection by phagocytosis is preponderant when infected T cells are dead or dying. This is consistent with the results from the Sattentau group reporting that macrophages engulf ~50-fold more dying infected T cells than healthy uninfected cells (Baxter et al., 2014). These two

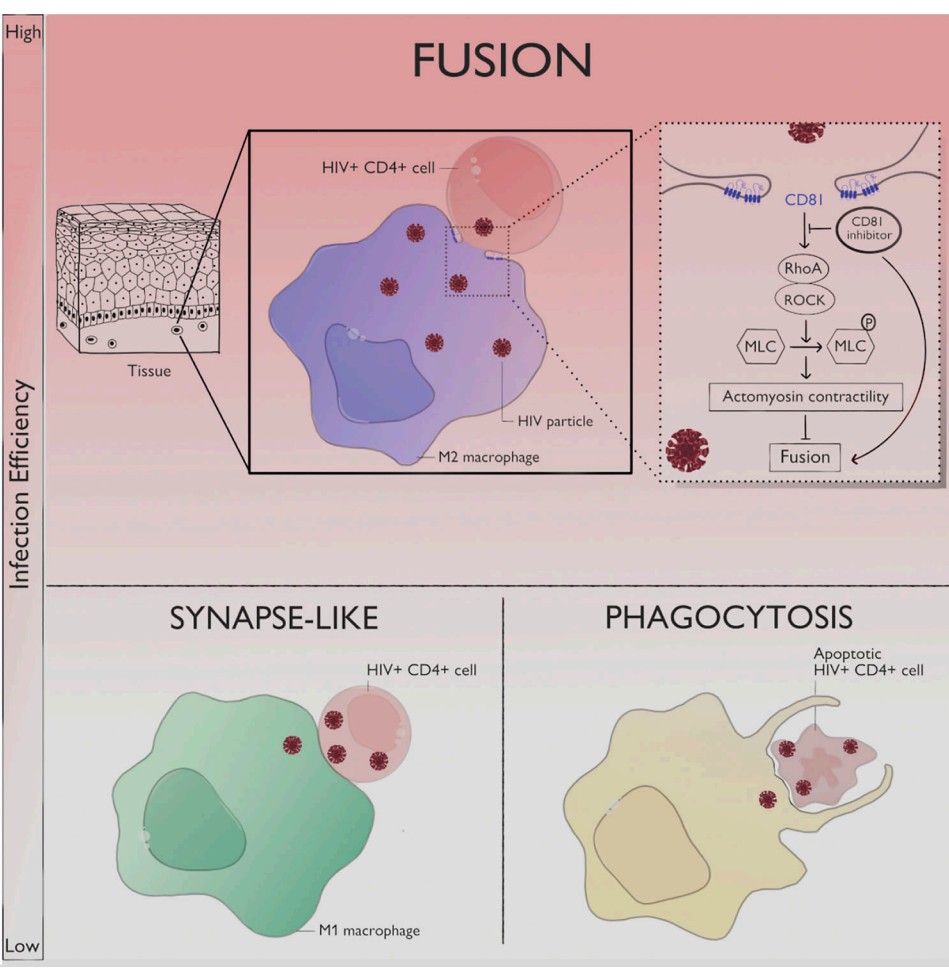

Figure 6.   **Mechanisms of cell-to-cell HIV-1 transfer toward macrophages.** Three mechanisms are involved in macrophage infection by virus transfer from infected CD4⁺ T cells and their relative contribution is influenced by the macrophage activation status and the state of viability of the infected T cells. The synapse mode of infection becomes predominant in M1 macrophages but results in low permissiveness of macrophages to HIV-1 infection. The phagocytosis-mediated mechanism happens primarily when HIV-1-infected T cells are apoptotic and is poorly efficient for macrophage infection. The most productive viral infection of macrophages is observed through heterotypic cell fusion with viable infected T cells. This process occurs in several tissue-resident or infiltrated macrophage types and is negatively regulated by the CD81/RhoA-ROCK/Myosin II axis. This viral transmission route is promoted by an anti-inflammatory environment. Illustration by Claire Lastrucci.

mechanisms described in vitro may explain the presence of T cell markers and viral DNA originating from infected CD4⁺ T cells in myeloid cells from lymphoid tissues of SIV-infected macaques (Calantone et al., 2014; DiNapoli et al., 2017). Indeed, we also detected the presence of T cell rearranged TCR DNA in MDMs infected upon fusion with infected Jurkat cells (unpublished data). Our results are also in agreement with the existence of MGCs derived from dendritic cells or macrophages in tissues even under effective suppressive cART (Hendricks et al., 2021). In addition, we give evidence for a third and not yet described mechanism of macrophage infection related to the virological synapse. We have previously proposed this synapse-like mechanism of infection for monocytes and dendritic cells (Xie et al., 2019). Here, we show that this mode of infection is characterized by a prolonged contact between infected T lymphocytes and macrophages followed by cell detachment and a lower infection efficiency compared to virus transfer by cell fusion (Fig. S6 A and Video 5). Whatever the living or dying state of infected

lymphocytes, the synapse-like viral transmission mode toward MDMs remains minor in an anti-inflammatory environment.

We also report that the activation status of macrophages is another critical parameter governing macrophage infection efficacy. Indeed, macrophage populations in tissues are highly heterogeneous, depending on their origin, anatomical location, and cytokine environment (Bleriot et al., 2020; Murray, 2017). Moreover, the activation state of macrophages varies during disease progression and is instrumental in modulating the efficacy of cell-free viral infection (Cassol et al., 2010). For example, in the endometrium of non-pregnant women and in decidua, resident macrophages have a mixed M1/M2 phenotype (Quillay et al., 2015). When maintained in culture, these cells switched to a more M2 phenotype and became more permissive to cell-free viral infection (El Costa et al., 2016b). In this study, we showed that pro-inflammatory MDMs display a low HIV-1 infection permissiveness corresponding to a strong decrease of infection via heterotypic cell-fusion with infected T cells, while the

synapse mode of infection became predominant. This could be explained, at least in part, by the reduction, in these M1 cells, of the cell surface expression of CD4 (this study and Cassol et al., 2009) which is necessary for infection by heterotypic cell fusion (Bracq et al., 2017). In conditions with living infected T cells, macrophage infection by phagocytosis (Baxter et al., 2014) was rare and we showed that it was weakly affected by the activation state. Altogether our data show that the three virus transmission routes for macrophage infection can coexist and that their relative contribution is dictated by the characteristics of both the donor T cells (its viability) and the target macrophages (its activation status; Fig. 6).

As the use human MDMs could not recapitulate the characteristics and complexity of tissue-resident macrophages and their environment, it was important to assess the potential infection by heterotypic cell fusion in terminally differentiated tissue macrophages. We observed that heterotypic cell fusion occurs with macrophages inside explants of synovial tissue and with several purified macrophages from placenta (Jabrane-Ferrat and El Costa, 2022), lymphoid tissues (i.e., tonsils; Glushakova et al., 1995), and lung, where alveolar macrophages represent HIV-1 reservoirs (Jambo et al., 2014). This is consistent with a recent study that showed that R5 T cell-tropic (T-tropic) viruses, mainly circulating during acute and chronic infections, can infect alveolar macrophages through contact with infected CD4+ T cells (Schiff et al., 2021). Thus, any type of tissue macrophages examined in this study can be infected by heterotypic fusion with infected T cells and some specific environments or disease stages could favor this infection mode.

During infection through heterotypic fusion, macrophages established specific contacts with infected T cells where β2 integrin and F-actin are instrumental. Although such features have been also described for the virological synapse formed between CD4+ T lymphocytes (Jolly et al., 2004; Jolly et al., 2007a; Jolly et al., 2007b; Jolly and Sattentau, 2007), the T cell-macrophage interactions presented a striking difference, namely, the triggering of cell fusion leading to the formation of infected MGCs. Interestingly, we found that CD81 has a negative effect on heterotypic fusion with infected T lymphocytes. It is known that CD81 participates, in concert with other tetraspanins, like CD63 and CD9, in many biological processes such as cell adhesion, migration, cell–cell fusion as well as in multiple steps of viral infection, especially in the case of HIV-1 (Ho et al., 2006; Jolly and Sattentau, 2007). In particular, CD81 has been shown to inhibit HIV-1-mediated syncytium formation in both infected and target cells, mainly using transformed cell lines as virus producers and T cells as targets (Gordon-Alonso et al., 2006; Weng et al., 2009). Our study, showing the restriction role of CD81 in heterotypic cell fusion, contributes to a better understanding of the role of tetraspanins in fusion processes. As tetraspanins are organized in membrane microdomains, CD81 could trap proteins involved in cell fusion, such as integrins among others (Dufrancais et al., 2021; Hemler, 2005). The large extracellular loop of CD81 could also interact with gp41 exposed at the membrane of infected T cells, limiting the heterotypic cell fusion, as described for CD63 in the virological synapse (Ivanusic

et al., 2021; Kitadokoro et al., 2001). Moreover, our data disclose additional mechanism by which CD81 inhibits HIV-1-mediated heterotypic fusion through RhoA/ROCK activation and subsequent phosphorylation of myosin IIA, thus disturbing the macrophage actomyosin network unfavorable to fusion (Fig. 6). Thus, we propose that, when a macrophage establishes contacts with an infected T cell, CD81 initiates a combination of events, which activate RhoA/ROCK with subsequent activation by phosphorylation of myosin IIA, resulting in a strong cortical network unfavorable to the cell fusion with infected T CD4+ cells. This could induce mechanical tensions at the membrane limiting cell adherence and probably enlargement of the gp41-mediated fusion pore necessary for membrane fusion between infected T cells and macrophages (Chen et al., 2008). However, macrophages have evolved specific strategies to overcome this restriction pathway to allow the formation of MGCs. Exploring the role of the other macrophage tetraspanins may be an avenue to explore to better understand how this limitation is bypassed. MGC formation in tissues being supportive for a sustained infection and associated with the severity of the disorders (Compton and Schwartz, 2017; Orenstein, 2000), controlling the size of MGCs could help for cell survival and thus maintenance of long-term infected foci.

Altogether our results strongly support that heterotypic fusion between infected CD4+ T cells and macrophages is a relevant and very efficient process for viral transmission to tissue macrophages. It remains to be determined in which tissue it indeed occurs in vivo, which is a huge challenge. One indication is that in chronically SIV-infected macaques, alveolar macrophages or infected tissue-resident macrophages from bowel or spleen contain evidence of CD4+ T cell material (Calantone et al., 2014; DiNapoli et al., 2016; DiNapoli et al., 2017). It is also interesting to question whether the resulting heterokaryon has a dominant macrophage phenotype, as already proposed (Martinez-Mendez et al., 2017), or whether it combines properties favorable to virus dissemination and persistence, such as the long-life status of infected macrophages and the capacities inherited from T cells to bypass viral restriction (Xie et al., 2019). Importantly, our findings can be transposed to infection of other myeloid cells as a similar heterotypic cell fusion mechanism is used to transfer the virus from infected T lymphocytes to immature dendritic cells and osteoclasts (Raynaud-Messina et al., 2018; Xie et al., 2019). In addition, although this transmission route does not allow for infection of macrophages with X4 tropic strains (Bracq et al., 2017), in contrast to cell-free viral infection, it allows for efficient transfer toward macrophages of HIV-1 strains defined as non-macrophage-tropic, such as T/F viruses involved in sexual transmission and early steps of virus dissemination (Han et al., 2022). This transmission through heterotypic cell fusion is associated with enhanced Env interactions with CD4 and the co-receptor CCR5, resulting in a viral entry less dependent on the expression level of both receptors on macrophages. In conclusion, our results not only give crucial information for how macrophages are infected by HIV-1 and participate in the associated pathogenesis but also could yield novel therapeutic strategies to directly interfere with viral cell-to-cell spreading and thus virus dissemination.

## Materials and methods

### Cell culture

HEK293T, TZM-bl, and Jurkat E6.1 cell lines were obtained through the NIH HIV Reagent Program, Division of AIDS, NIAID, NIH, contributed by Dr. Andrew Rice, Dr. John C. Kappes, Dr. Xiaoyun Wu, Tranzyme Inc., and ATCC (Dr. Arthur Weiss). HEK293T and TZM-bl cells were maintained in Dulbecco Minimal Essential Medium (DMEM, Gibco) and Jurkat were maintained in Roswell Park Memorial Institute medium (RPMI 1640; Gibco). All culture media were supplemented with 10% Fetal Bovine Serum (FBS, Sigma-Aldrich), L-glutamin (10 mM; Gibco), and penicillin-streptomycin (1%; Gibco). Human primary monocytes were isolated from healthy subject (HS) buffy coat (provided by Etablissement Français du Sang, Toulouse, France, under contract 21/PLER/TOU/IPBS01/20,130,042) and differentiated toward macrophages (Souriant et al., 2019). According to articles L12434 and R124361 of the French Public Health Code, the contract was approved by the French Ministry of Science and Technology (agreement number AC 2009921). Written informed consents were obtained from the donors before sample collection. Briefly, peripheral blood mononuclear cells (PBMCs) were recovered by gradient centrifugation on Ficoll-Paque Plus (GE Healthcare). CD14[+] monocytes were then isolated by positive selection magnetic sorting, using human CD14 Microbeads and LS columns (Miltenyi Biotec). Cells were then seeded on glass coverslips at $1.10^6$ or $5.10^5$ cells per well in 12- or 24-well plates, respectively, and allowed to differentiate for 5–7 d in RPMI-1640 medium supplemented with 10% FBS, L-glutamin, penicillin-streptomycin, and recombinant human M-CSF (20 ng/ml; Peprotech) or GM-CSF (25 ng/ml; Miltenyi Biotec). For polarization experiments, macrophages were washed twice after 5 d of differentiation and exposed to either a combination of *E. coli* lipopolysaccharide (LPS, 10 ng/ml; BioXtra) and interferon-g (IFNg, 10 ng/ml; Miltenyi Biotec), interleukin-4 (IL-4, 20 ng/ml; Miltenyi Biotec), or interleukin-10 (IL-10, 20 ng/ml; Miltenyi Biotec) for 48 h. Human primary CD4[+] T cells were isolated and purified (Bracq et al., 2017). Briefly, the CD4[+] T cells were purified from PBMCs by positive selection (CD4[+] T cell isolation kit; Miltenyi). CD4[+] T cells were activated for 3 d in RPMI 1640 medium containing 20% fetal bovine serum (FBS), interleukin-2 (IL-2; Miltenyi) at 10 U/ml, and phytohemagglutinin-P (PHA-P; Sigma-Aldrich) at 5 µg/ml. After activation, CD4[+] T cells were kept in RPMI 1640 medium supplemented with 20% FBS and IL-2. All cells were grown at 37°C under 5% $CO_2$.

### Viruses

Virus stocks were generated by transient co-transfection of HEK293T cells with proviral plasmids coding for HIV-1 NLAd8 or ADA-GFP and the same viruses with VSV-G envelope as previously described (Souriant et al., 2019). Supernatant were harvested 72-h post-transfection, centrifuged to remove cell debris and stocked at –80°C. The infectivity of the virus stock and culture supernatant was assessed by titration on Jurkat and TZM-bl cell lines and quantified by flow cytometry or fluorescence microscopy analysis, respectively. HIV-1 infectious units were quantified by IF, as reported (Souriant et al., 2019; Verollet et al., 2015) using TZM-bl reporter cells. TZM-bl cells are Hela cell lines (female origin) that were obtained through NIH AIDS Reagent Program, Division of AIDS, NIAID, NIH from Dr. John C. Kappes, Dr. Xiaoyun Wu, and Tranzyme Inc.).

### HIV-1 infection of CD4[+] T cell and co-cultures with macrophages

Jurkat cells were adjusted at $5.10^5$ cells/ml and infected as previously described (Bracq et al., 2017). Cells were exposed to viral stock at a M.O.I. of 0.2 for 24 h, washed twice with phosphate buffer saline (PBS, Gibco) to remove remaining pseudotyped virus, and incubated for 24 h or 7 d more before co-culture. Infected Jurkat cells were washed again twice with PBS, adjusted at a density of $2.10^6$ cells/ml in complete RPMI, and added to macrophage at a ratio of 2:1. For some experiments, infected Jurkat cells were stained with CellTracker Blue CMAC and/or pHrodo Red succinimidyl ester (Invitrogen), and macrophages were stained with LysoTracker Red DND-99 (Invitrogen) or Celltracker Green CMFDA and Violet BMQC (Invitrogen) before co-culture. After 24 h of co-culture, Jurkat cells were removed by washing 3 times with PBS and macrophage infection was quantified by either flow cytometry or fluorescence microscopy analysis. In some experiments, macrophages were kept in culture for 5 d after washing Jurkat cells, and the level of HIV-p24 released in the culture medium was quantified by homemade sandwich ELISA (Dupont et al., 2020; Souriant et al., 2019) with reagents obtained through the NIH HIV Reagent Program, Division of AIDS, NIAID, NIH, contributed by Dr. Bruce Chesebro, Kathy Wehrly, NABI, and National Heart Lung and Blood Institute (Dr. Luiz Barbosa).

### Chemical inhibitors and blocking antibodies

For blocking experiments, macrophages were first incubated for 1 h in the presence of Human Trustain FcX (1:50; BioLegend) diluted in complete RPMI to prevent the binding of blocking antibodies to Fc receptors expressed by macrophages. The culture medium was then removed and replaced by a dilution of anti-CD18 (clone TS1/18, mouse IgG1, 302102; Biolegend), anti-CD81 (clone 5A6, mouse IgG1, 349502; Biolegend), anti-CD9 (clone M-L13, mouse IgG1, 555370; BD Biosciences), anti-CD4 (clone Leu3a, mouse IgG1, 344602; Biolegend), or mouse IgG1k isotype control (clone MOPC-21, mouse IgG1, 400102; Biolegend). All antibodies were purchased from BioLegend and used at 10 µg/ml. After 1 h of incubation with antibodies, infected T cells were directly added to the medium. Inhibitors of actin remodeling were Jasplakinolide (500 nM; Sigma-Aldrich), cytochalasin D (1 µM; Sigma-Aldrich), and CK666 (100 µM; Abcam). Inhibitors of Rho/ROCK/MyosinII were blebbistatin (25 µM; Sigma-Aldrich), Y27632 (50 µM; Sigma-Aldrich), and C3 exoenzyme coupled to permeant peptide TAT (TAT-C3, 10 µg/ml) that was produced and purified in G. Fabre's laboratory (Sahai and Olson, 2006). The phosphatase inhibitor Calyculin A (Sigma-Aldrich) was used at 5 nM. All chemical inhibitors were used 1 h before starting co-cultures, except for TAT-C3 which was added 48 h before. All blocking antibodies and inhibitors were kept during the co-culture.

### Tissue processing and resident macrophages isolation

Human tonsillar tissues were collected after a tonsillectomy procedure in children with tonsil hyperplasy leading to

breathing disorders and processed (Smith et al., 2020). Samples were used in accordance with the French Ministry for Higher Education and Research (certification number #DC-2019-3370) and after informed consent from the parents. Briefly, tissues were cut into fragments within PBS supplemented with 5% FBS, L-glutamine (10 mM), Penicillin (100 units/ml), Streptomycin (100 µg/ml), and Gentamicin (50 µg/ml). The tissue fragments were then pressed through a cell strainer (40 µm) and the resuspended cells were collected within the supplemented PBS. All tissue processing steps were carried out on ice. Mononuclear cells were then recovered by gradient centrifugation on Ficoll-Paque, and CD14$^+$ cells were isolated by positive selection magnetic sorting before seeding in 24-well plates. After 2 h, non-adherent cells were removed by washing before co-culture experiments.

The study concerning first trimester human placenta was approved by the Research Ethical Comity Haute-Garonne and Agence de Biomédecine (PFS08-022). Hofbauer cells were purified by CD14 positive selection kit according to the manufacturer procedure (Miltenyi Biotec; Jabrane-Ferrat and Veas, 2020). Cell purity was at least 90% as determined by CD14 and CD68 staining. For exposition of Hofbauer cells to cell-free virus, the supernatant of 24-h-infected Jurkat cells was collected, filtered (0.2 µm pores), and a volume corresponding to the number of Jurkat cells needed for co-culture was added to macrophages.

Broncho-alveolar lavage was performed using a flexible bronchoscope inserted through the nose under local anesthesia. The patient gave her informed consent for the use of the leftover of the sample for research purpose. Human alveolar macrophages were collected from broncho-alveolar lavage and seeded on glass coverslides for 1 h before washing and kept in RPMI 10% FBS (Sigma-Aldrich), L-glutamin (10 mM; Gibco) and penicillin-streptomycin (1%; Gibco). According to French law on ethics, patient was informed that his codified sample's leftover, coming from the biological collection from Toulouse Universitary Hospital registered by the French Ministry of Higher Education and Research under the number DC-2020-4074, will be used for this research purpose.

Synovial tissues obtained from arthroplastic surgery of nonarthritic patients were cut into blocks of around 1–2 mm³ and incubated on gel-foam sponges (Pfizer) with M-CSF (50 ng/ml). Explants were then exposed to HIV-1 infected Jurkat cells for 24 h, washed and maintained in culture for 24–72 h in α-MEM supplemented with 10% FBS and M-CSF (50 ng/ml). Tissues were then fixed in 10% phosphate-buffered formalin and embedded in paraffin as in Raynaud-Messina et al. (2018). For immunohistofluorescence of paraffin sections (5 µm), heat-induced epitope retrieval was performed in 10 mM sodium-citrate (pH 6.0) and stained ON at 4°C with antibodies to p24 (Kal1, mouse IgG, M0857, 1:10; DAKO), CD3 (SP7, rabbit, GTX16669, 1:100; Genetex), and CD68 Alexa Fluor 647 (KP1, mouse IgG1, sc-20060 AF647, 1:50; Santacruz). Goat anti-mouse Alexa Fluor 488 (#4408, 1:400; Cell signaling), Goat anti-rabbit Alexa Fluor 555 (#4413, 1:400; Cell signaling) secondary antibodies or isotype control mouse IgG1 Alexa Fluor 647 (MOPC21,

1:50; BioLegend) were used to detect the corresponding primary antibodies. Nuclei were visualized with 4′,6-diaminidino-2-phenylindole (DAPI). Images were acquired using a Zeiss Axio Imager M2 using an X40/0.95 Plan Apochromat objective (Zeiss) and processed using the Zeiss Zen software ORCA-flash 4.0 LT (Hamamatsu) camera.

## siRNA silencing

Macrophages were transfected after 3 d of differentiation with 166 nM siRNA control or targeting CD81 (siCTL or siCD81), or a combination of siRNA targeting MYH9 and MYH10 (siMYO) at 83 nM each using the HiPerfect system (Qiagen; Bouissou et al., 2017; Troegeler et al., 2017). The mix of HiPerfect and siRNA was incubated for 15 min at room temperature and then added drop by drop to the cells. After 6 h of transfection, cells were incubated for 6 d in complete RPMI with M-CSF (20 ng/ml). Target protein expression was quantified by flow cytometry for CD81 or Western blot analysis for Myosin II. An approximate 50% protein depletion was obtained for both targets (see Fig. S5). The following siRNA (Dharmacon, Horizon Discovery) were used.

- human ON-TARGET plus SMART pool siRNA nontargeting control pool (siCtrl): 5′-UGGUUUACAUGUCGACUAA-3′
- ON-TARGETplus Human CD81 siRNA - SMARTpool (siCD81): 5′-GAACAGCUCCGUGUACUGA-3′; 5′-GCCCAACACCUUCUAUGUA-3′; 5′-GUGGAGGGCUGCACCAAGU-3′; 5′-CCAACAACGCCAAGGCUGU-3′
- ON-TARGETplus Human MYH9 siRNA—SMARTpool: 5′-GUAUCAAUGUGACCGAUUU-3′; 5′-CAAAGGAGCCCUGGCGUUA-3′; 5′-GGAGGAACGCCGAGCAGUA-3′; 5′-CGAAGCGGGUGAAAGCAAA-3′
- ON-TARGETplus Human MYH10 siRNA—SMARTpool: 5′-GGAAGAAGCUCGACGCGCA-3′; 5′-GAGCAGCCGCCAACAAAUU-3′; 5′-GGGCAACUCUACAAAGAAU-3′; 5′-CCAAUUUACUCUGAGAAUA-3′.

## Western blot analysis

Cells were washed wish PBS and lysed with cold PBS supplemented with Triton X-100 (1%; Sigma-Aldrich), ethylenediaminetetraacetic acid (EDTA, 5 mM) and protease and phosphatase inhibitors (Halt Protease Inhibitor Cocktail, Thermo Fisher Scientific). Total protein concentration was determined in lysates with the BCA Protein Assay (Thermo Fisher Scientific). 20 µg of proteins were mixed with Bolt LDS Sample Buffer and Sample Reducing Agent (Invitrogen), heated at 70°C and loaded in 4–12% Bis-Tris Mini Protein Gels (Invitrogen), subjected to electrophoresis and transferred onto nitrocellulose membranes. Membranes were then saturated with 3% bovine serum albumin (BSA, Euromedex) in Tris-buffered saline with Tween-20 (TBS-T, 0.1%) for 1 h and incubated overnight at 4°C with anti-actin (1:10,000, rabbit, A5060; Sigma-Aldrich), anti-Myosin IIA (1:500, rabbit, 909801; BioLegend), anti-(p-Ser19)-MLC (1:500, rabbit, 3671S; Cell Signaling), or anti-RhoA (1:500, mouse IgG1, sc-418; Santa Cruz Biotechnology) diluted in saturation buffer. Membranes were incubated with a horseradish peroxidase-coupled goat anti-rabbit (1:10,000, P0448; Dako) secondary antibody for 1 h and horseradish peroxidase

activity was revealed using a chemiluminescence kit (Cytiva). Chemiluminescence was detected with ChemiDoc Touch Imaging System (Bio-Rad Laboratories). Quantification of immunoblot intensity was performed using Image Lab (Bio-Rad) and the intensity of each band is normalized to actin.

## Immunofluorescence and microscopy
Cells were fixed with paraformaldehyde (PFA, 3.7%; Sigma-Aldrich), sucrose 30 mM in PBS (Gibco), permeabilized with Triton X-100 0.3% (Sigma-Aldrich) for 10 min, and saturated with BSA (1% in PBS) for 30 min. Coverslips were incubated with anti-HIV p24 (KC57-FITC, clone FH190-1-1, mouse IgG1, 1:100, 6604665; Beckman Coulter) or anti-CD18 (clone TS1/18, mouse IgG1, 1:100, 302102; Biolegend) or anti-MyosinIIA (rabbit, 1:100, 909801; Biolegend) diluted in saturation buffer for 1 h, washed with PBS and incubated with goat anti-mouse (4408s) or anti-rabbit (4412S) secondary antibody coupled to AlexaFluor-488 (2 μg/ml; Cell Signaling Technology), TexasRed-labeled phalloidin (Invitrogen) and either DAPI (500 ng/ml; Sigma-Aldrich) or DRAQ5 (2 μM; Thermo Fisher Scientific) for 30 min. After several washes with PBS, coverslips were mounted on a glass slide using fluorescence mounting medium (Dako) and stored at 4°C. For CD18 staining, incubation with primary antibody (TS1/18, 5 μg/ml; BioLegend) was performed on ice before fixation in culture medium. Images for quantification of infection were acquired using a Zeiss Axio Imager M2 and a 20×/0.8 Plan Apochromat or 40×/0.95 Plan Apochromat objectives (Zeiss). Images were acquired and processed using the Zeiss Zen software ORCA-flash 4.0 LT (Hamamatsu) camera. For confocal images, specimens were observed with a Zeiss LSM710 confocal microscope that uses a Zeiss AXIO Observer Z1 inverted microscope stand with transmitted (HAL), UV (HBO), and laser illumination sources. Images were acquired with a Zeiss ×63 (oil) NA 1.35 objective. Images in Fig. S4 B were obtained with an Elyra 7 lattice SIM microscope and laser illumination sources (Zeiss). For live imaging, specimens were observed on an Andor/Olympus spinning disk microscope equipped with a Yokogawa CSU-X1 scanner unit and an emCCD camera (Andor iXon 888) under the control of the iQ3 software (Andor Oxford Instruments company). Images were acquired with an Olympus ×60 (oil) NA 1.35 objective for DIC and GFP (HIV) signal. For all images, visualization and analysis was performed with ImageJ. The HIV infection index (total number of nuclei in HIV-stained cells divided by total number of nuclei × 100) was quantified, as in Souriant et al. (2019). F-actin and CD18 mean signal intensity was measured at the interface between T cells and macrophages on a single z-stack and was normalized to the sum of the mean signals measured in non-relevant parts of the membranes of both cells to generate a signal enrichment value. The fusion index is defined as the number of nuclei present in a multinucleated giant cell (>2 nuclei) relative to the total number of nuclei (Raynaud-Messina et al., 2018; Verollet et al., 2015).

## Flow cytometry
For analysis of macrophage infection after co-culture, remaining Jurkat cells were removed by washing three times with PBS and macrophages were incubated with Trypsin 0.05% EDTA (Gibco)

for 10 min at 37°C. Cells were then harvested by flushing thoroughly with cold staining buffer (PBS, 2 mM EDTA, 0.5% FBS). After 5 min centrifugation at 320 g, pellets were resuspended in cold staining buffer with PE-coupled anti-CD11b (M1/70, 1 μg/ml; BioLegend) and incubated on ice for 20 min. Cells were then washed twice with staining buffer and fixed with 3.7% PFA for 30 min. After fixation, cells were washed and stained with FITC-coupled anti-HIV p24 (KC57-FITC, 1:100; Beckman Coulter) in permeabilization buffer (PBS, 0.15% Triton X-100, 1% BSA) for 20 min. Finally, macrophages were washed twice, resuspended in PBS and analyzed by flow cytometry using BD LSR Fortessa (BD Biosciences) and the associated BD FACSDiva software. Macrophage infection was then processed using the FlowJo_V10 software by gating on macrophages according to their Size Scatter (SSC) properties and CD11b expression before doublet exclusion and analysis of their HIV p24 staining. We verified that the majority of selected HIV-p24+ macrophages were also positive for the T cell marker CD3, as previously reported (Bracq et al., 2018) and predominantly multinucleated after sorting (not shown). In addition, in most experiments, T20 and anti-CD4 blocking antibodies were used as controls as in Bracq et al. (2018). To determine surface protein expression, macrophages were detached as described above, incubated with Human Trustain FcX (1:100; BioLegend) and LIVE/DEAD Fixable Blue Dead Cell Stain (1:1,000; Molecular Probes) in cold staining buffer for 20 min, washed twice, and stained with AF488-coupled anti-CD4 (clone OKT4, mouse IgG2b, 317420; Biolegend), PE/Dazzle 594-coupled anti-CCR5 (clone J418F1, rat IgG2b, 359125; Biolegend), CD3-AlexaFluor647 (clone UCHT1, mouse IgG1, 300416; Biolegend), PE-coupled anti-CD11b (clone M1/70, rat IgG2b, 101208; Biolegend), FITC-coupled anti-HIV p24 (clone KC57/FH190-1-1, mouse IgG1, 6604665; Beckman Coulter) or the corresponding isotype controls in cold staining buffer for 20 min. All antibodies were used at 5 μg/ml and were purchased from BioLegend. Cells were then washed twice with cold staining buffer and directly analyzed by flow cytometry as described above. The expression of proteins of interest was determined by subtracting the Median Fluorescence Intensity (MFI) of the isotype control to each staining after exclusion of dead cells and doublets.

To analyze the viability of infected Jurkat cells, cells were washed and stained with LIVE/DEAD Fixable Blue Dead Cell Stain and APC-coupled Annexin V (1:100; BD Biosciences) in cold 1× Annexin V Binding Buffer (BD Biosciences) for 20 min. Cells were then washed, fixed, and stained for HIV p24.

## Scanning electron microscopy
Jurkat cells were infected with ADA-GFP-VSVG as described above. After 48 h, GFP positive cells were sorted with a FACS-Aria Fusion (BD Biosciences) and co-cultured with macrophages. After 30 min, the culture medium was discarded, and cells were fixed using 0.1 M sodium cacocylate buffer supplemented with 2.5% (vol/vol) glutaraldehyde. Then, they were washed three times for 5 min in 0.2 M cacodylate buffer (pH 7.4), post-fixed for 1 h in 1% (wt/vol) osmium tetroxide in 0.2 M cacodylate buffer (pH 7.4), and washed with distilled water. Samples were dehydrated through a graded series (25–100%) of ethanol,

transferred in acetone, and subjected to critical point drying with $CO_2$ in a Leica EM CPD300. Dried specimens were sputter-coated with 3 nm platinum with a Leica EM MED020 evaporator and were examined and photographed with a FEI Quanta FEG250.

### Quantification of RHOA activation

Pull-down of activated RHOA was performed as previously (Bery et al., 2019). Cells ($2.10^6$ per condition) were lysed in buffer (50 mM Tris, pH 7.4, 500 mM NaCl/10 mM MgCl 2/0.5% Triton X-100 supplemented with protease and phosphatase inhibitors). GST-RBD (30 mg) was incubated with cleared lysate for 45 min at 4°C, then beads were washed three times with washing buffer (50 mM Tris-HCl, pH 7.5, 150 mM NaCl, 10 mM MgCl 2, 0.1% Tween20) and denatured in 2× Laemmli reducing sample buffer, boiled for 5 min and separated on 12.5% SDS-PAGE for Western Blot analysis.

### Quantification and statistical analysis

All statistical analyses were performed using GraphPad Prism 9 (GraphPad Software Inc.). Two-tailed paired or unpaired $t$ test was applied on data sets with a normal distribution (determined using Kolmogorov–Smirnov test), whereas two-tailed Mann–Whitney (unpaired test) or Wilcoxon matched-paired signed rank tests were used otherwise. Bar histograms represent mean with standard deviation for data with a normal distribution, and median with interquartile range otherwise. When multiple comparisons were performed, the statistical analyses used were detailed in the corresponding figure legend. $P < 0.05$ was considered as the level of statistical significance (*, $P \leq 0.05$; **, $P \leq 0.01$; *** $P \leq 0.001$; **** $P \leq 0.0001$).

### Online supplemental material

Fig. S1 (in support of Fig. 1) shows cell viability and infection rate of Jurkat T cells. It also shows the three mechanisms of HIV-1 transfer from infected Jurkat cells to MDMs. Fig. S2 (in support of Fig. 1) shows that most infected MDMs contain T material, and that MDM infection with 2-d-infected Jurkat cells is more efficient than MDM infection with 7-d-infected Jurkat cells. Fig. S3 (in support of Fig. 2) shows that human tissue macrophages (from synovial explants, A and broncho-alveolar lavages, B) are infected by fusion with infected Jurkat T cells. It also shows that CD4 is necessary for fusion of infected Jurkat cells with alveolar macrophages. Fig. S4 (in support of Fig. 2) shows human macrophages from placenta are infected by fusion with infected Jurkat T cells in a CD4-dependent manner. Fig. S5 (in support of Fig. 3) shows that MDM polarization regulates their ability to fuse with infected Jurkat cells or infected primary human CD4+ T lymphocytes. It also shows that CD4 expression level at the surface varies depending on the MDM activation state. Fig. S6 (in support of Fig. 4) shows images from Video 5: a synaptic contact between infected Jurkat T cells and MDMs. It also shows the effects of drug treatments on the actomyosin cytoskeleton of MDMs. Fig. S7 (in support of Figs. 4 and 5) shows siRNA efficiency and the effects of siRNA treatment (against Myosin IIA and CD81) on the actomyosin cytoskeleton of MDMs. It also

shows that inhibition of CD81 increases the fusion of infected Jurkat cells with alveolar macrophages.

### Data availability

Images of uncropped blots are available as source data files.

## Acknowledgments

We greatly acknowledge Myriam Ben Neji, Flavie Moreau, and Céline Berrone, IPBS and Genotoul Anexplo-IPBS, for accessing the BSL3 facilities; and the Genotoul TRI-IPBS facilities for imaging and flow cytometry, in particular Emmanuelle Näser, Serge Mazères, and Antonio Peixoto, and Raphaëlle Romieu-Mourez (Human immune-monitoring core, Infinity, Toulouse) for her help. We also thank Etienne Cavaignac for providing synovium samples and Aurélien Brindel for the BAL swabs. We greatly thank Loïc Rolas, Loïc Dupré, and Raphael Gaudin for fruitful discussions and Claire Lastrucci for illustration of Fig. 6.

This work was supported by the Centre National de la Recherche Scientifique, Université Paul Sabatier, the Institut National de la Santé et de la Recherche Médicale, the Agence Nationale de la Recherche (ANR16-CE13-0005-01, ANR DFG 2020 JA-3038/2-1), the Agence Nationale de Recherche sur le Sida et les hépatites virales (ANRS) grant nos. ANRS2018-02, ECTZ 118551/118554, ECTZ 205320/305352, ANRS ECTZ103104, and ECTZ101971, Sidaction (13457), the Fondation pour la Recherche Médicale (DEQ2016 0334894, DEQ2016 0334902, ENV202003011510), the Fondation Toulouse Cancer Santé and l'INSERM Plan Cancer. We also thank the AIDS Research and Reference Reagent Program, Division of AIDS, NIAID. R. Mascarau and V. Cantaloube-Ferrieu are PhD candidates at Université Toulouse III Paul Sabatier. M. Woottum is a PhD candidate at Université Paris Cité. R. Mascarau was supported by doctoral scholarships from Toulouse Cancer Santé, ANRS, and the Fondation des Treilles. M. Woottum and V. Cantaloube-Ferrieu were supported by doctoral scholarships from ANRS.

Author contributions: Conceptualization: R. Mascarau, B. Raynaud-Messina, and C. Vérollet; Methodology: R. Mascarau, Z. Valhas, R. Gence, N. Jabrane-Ferrat, J.-L. Davignon, B. Lagane, R. Poincloux, S. Bénichou, B. Raynaud-Messina, and C. Vérollet; Formal Analysis: R. Mascarau; Investigation: R. Mascarau, M. Woottum, L. Fromont, Z. Valhas, F. Bertrand, T. Beunon, A. Métais, V. Cantaloube-Ferrieu, B. Raynaud-Messina, and C. Vérollet; Resources: H.El Costa, N. Jabrane-Ferrat, Y. Gallois, N. Guibert, J-L. Davignon, G. Favre, B. Lagane, and S. Bénichou; Writing—Original Draft: R. Mascarau, B. Raynaud-Messina, and C. Vérollet; Writing—Review & Editing: R. Mascarau, R. Gence, A. Métais, N. Jabrane-Ferrat, N. Guibert, B. Lagane, I. Maridonneau-Parini, R. Poincloux, S. Bénichou, B. Raynaud-Messina, and C. Vérollet; Visualization: R. Mascarau, A. Métais, and R. Poincloux; Supervision: N. Jabrane-Ferrat, G. Favre, B. Lagane, S. Bénichou, B. Raynaud-Messina, and C. Vérollet; Project Administration: B. Raynaud-Messina and C. Vérollet; Funding Acquisition: I. Maridonneau-Parini, R. Poincloux, B. Lagane, S. Bénichou, and C. Vérollet.

Disclosures: The authors declare no competing interests exist.

Submitted: 20 May 2022

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

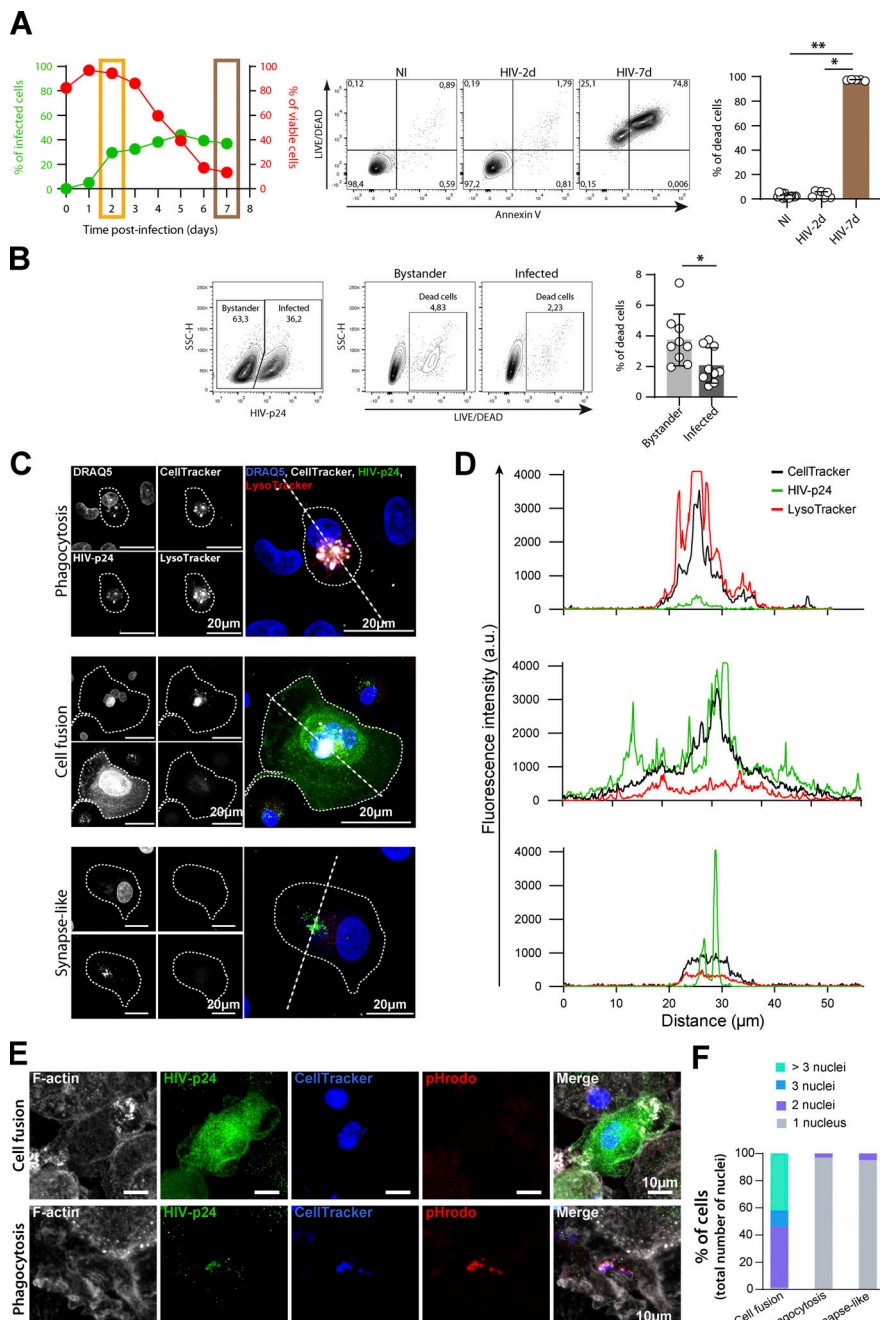

Figure S1. **Related to** Fig. 1. **(A)** Flow cytometry analysis of cell viability and infection rate of uninfected Jurkat cells (NI) or Jurkat cells infected with NLAD8-VSVG for 2 d (HIV-2d) or 7 d (HIV-7d). Left: Representative kinetics of Jurkat cell viability (%, red) and infection rate (% of HIV-p24+, green); frames indicate the time post-infection (2 d, yellow; 7 d, brown) selected for the rest of the study. Middle: Representative dot plot of Annexin V and LIVE/DEAD signals and gating strategy for selection of dead cells (LIVE/DEAD positive). Right: Quantification of % of dead cells (n = 6 donors, median ± interquartile range). **(B)** Flow cytometry analysis of 2 d-infected Jurkat cell viability in infected cells versus non-infected bystander cells. Left: Representative dot plot of HIV-p24 staining and gating strategy for selection of bystander and infected cells. Middle: Representative dot plots of LIVE/DEAD signals and gating strategy for selection of dead cells. Right: Quantification of % of dead cells in bystander and infected cells (n = 9 donors, mean ± SD). **(C–F)** Analysis of the three mechanisms of HIV-1 transfer from infected Jurkat cells to MDMs. **(C and D)** MDMs loaded with LysoTracker (red) were co-cultured with Jurkat cells infected with HIV-1 for 2 or 7 d and pre-loaded with CellTracker (gray). **(C)** Representative images of MDMs infected by phagocytosis (top), cell fusion (middle), or a synapse-like mechanism (bottom); HIV-p24 in green and nuclei in blue (DRAQ5). Infected MDMs were delineated by dotted lines. Scale bars, 20 μm. **(D)** Fluorescence intensity of HIV-p24 (green), LysoTracker (red), and CellTracker (black) signals along the dashed straight lines from right panels in C. **(E)** MDMs were co-cultured with Jurkat cells infected with HIV-1 (for 2 or 7 d) and pre-loaded with Celltracker (blue) and pHrodo (red). Representative images of MDMs infected by cell fusion (top) or phagocytosis (bottom). HIV-p24 in green and F-actin (phalloidin) in gray. Scale bars, 10 μm. **(F)** Quantification of the number of nuclei (DRAQ5+) in HIV-p24 + MDMs, from at least 200 cells per conditions. Quantification of a representative experiment is shown, from three experiments. **(E)** Flow cytometry analysis of MDMs co-cultured with uninfected (NI) or 2 d-infected Jurkat cells (HIV). Left: Representative dot plots of HIV-p24 and CD3 stainings and gating strategy to determine the % of CD3+ among HIV-p24 + MDMs. Right: Quantification. Statistical analyses: (A) Kruskal–Wallis multiple comparison test and (B) Paired t test. * P ≤ 0.05; ** P ≤ 0.01.

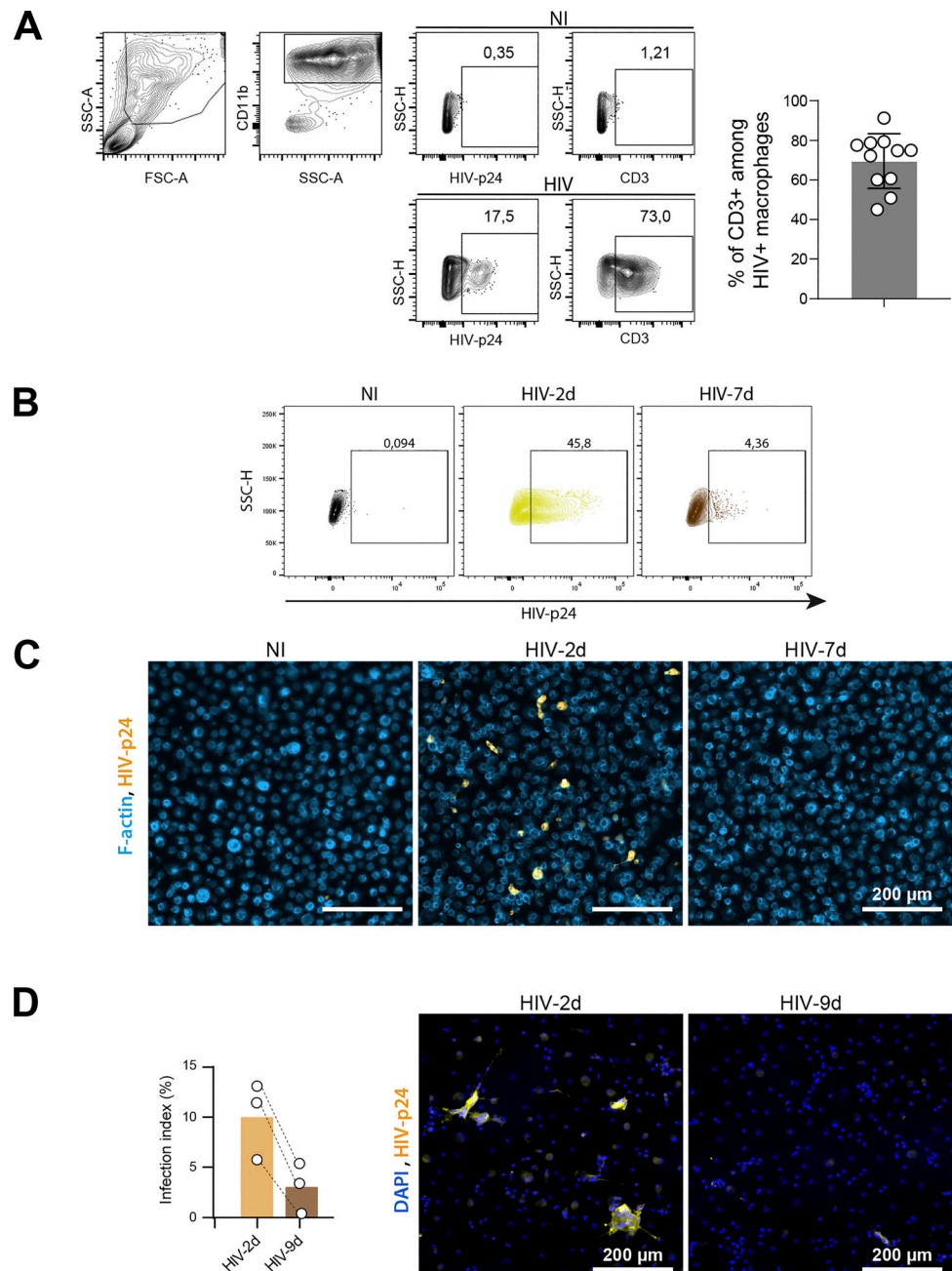

Figure S2. **Related to** Fig. 1. **(A)** Flow cytometry analysis of MDMs co-cultured with uninfected (NI) or 2 d-infected Jurkat cells (HIV). Left: Representative dot plots of HIV-p24 and CD3 stainings and gating strategy to determine the % of CD3+ among HIV-p24 + MDMs. Right: Quantification ($n$ = 11 donors). **(B and C)** MDM infection after a 24 h-co-culture with uninfected Jurkat cells (NI) or Jurkat cells infected for 2 d (HIV-2d) or 7 d (HIV-7d). **(B)** Flow cytometry analysis: Representative dot plots of HIV-p24 signals and gating strategy for selection of infected cells. See quantification in Fig. 1 D. **(C)** Representative images of microscopy analysis. F-actin (phalloidin, blue) and HIV-p24 (yellow). Scale bars, 200 μm. See quantification in Fig. 1 E. **(D)** Left: Quantification of MDM infection after a 24 h-co-culture with autologous CD4+ T cells infected for 2 d (HIV-2d) or 9 d (HIV-9d) ($n$ = 3 donors). Right: Representative images of microscopy analysis of nuclei (DAPI, blue) and HIV-p24 (yellow). Scale bars, 200 μm.

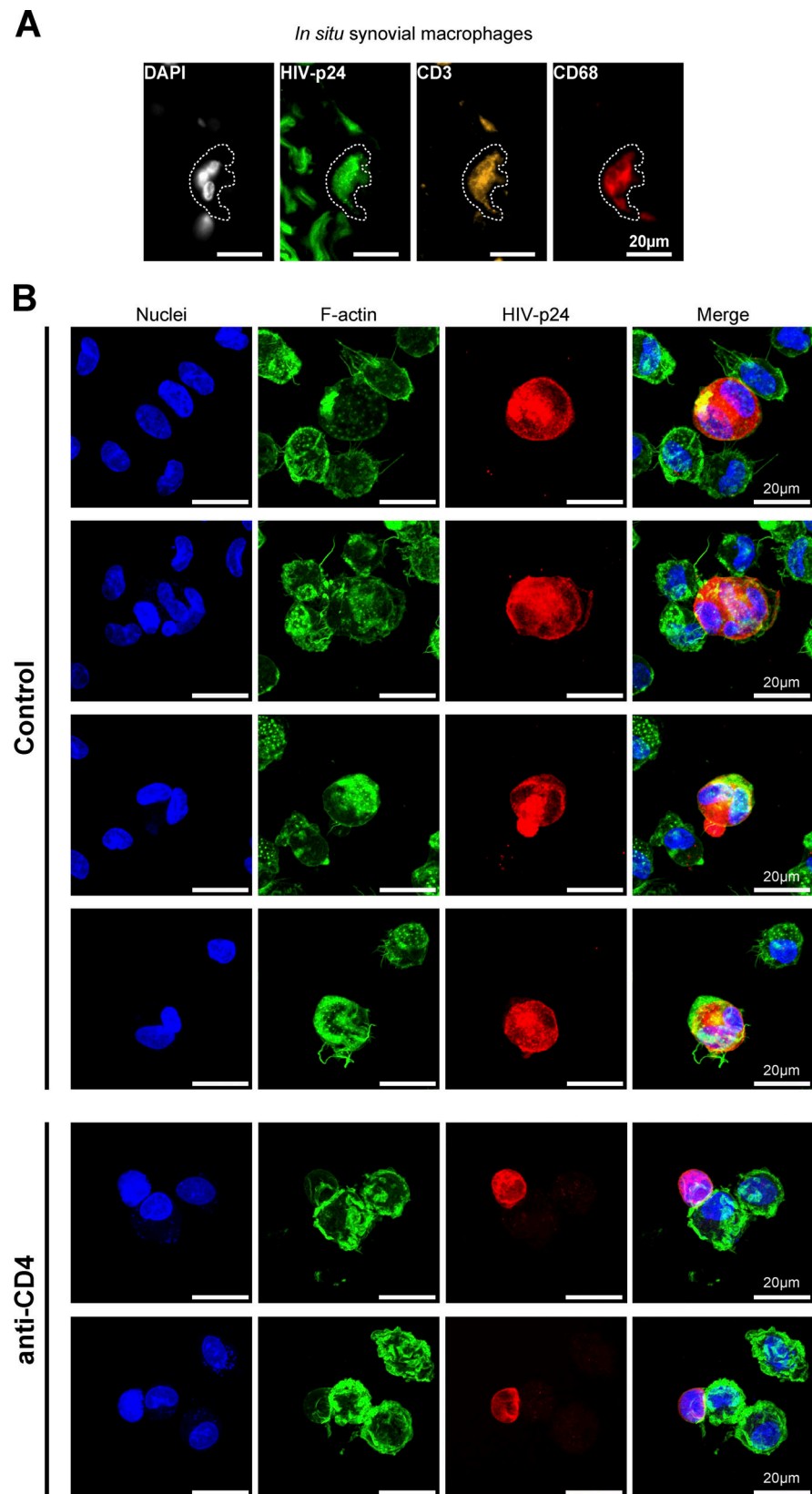

Figure S3. **Related to** Fig. 2. **(A)** Representative images of human synovial explants incubated with infected Jurkat cells during 24 h and processed for immunohistological analyses. Nuclei (DAPI, gray), HIV-p24 (green), CD3 (CD4$^+$ T cell marker, yellow), and CD68 (macrophage marker, red). Scale bar, 20 µm. **(B)** Gallery of representative images of ex vivo infection of alveolar macrophages isolated from adult broncho-alveolar lavages co-cultured for 24 h with 2 d-infected Jurkat cells in presence (anti-CD4) or not (Control) of neutralizing anti-CD4 antibody. Nuclei (DAPI, blue), F-actin (phalloidin, green), and HIV-p24 (red). Scale bars, 20 µm.

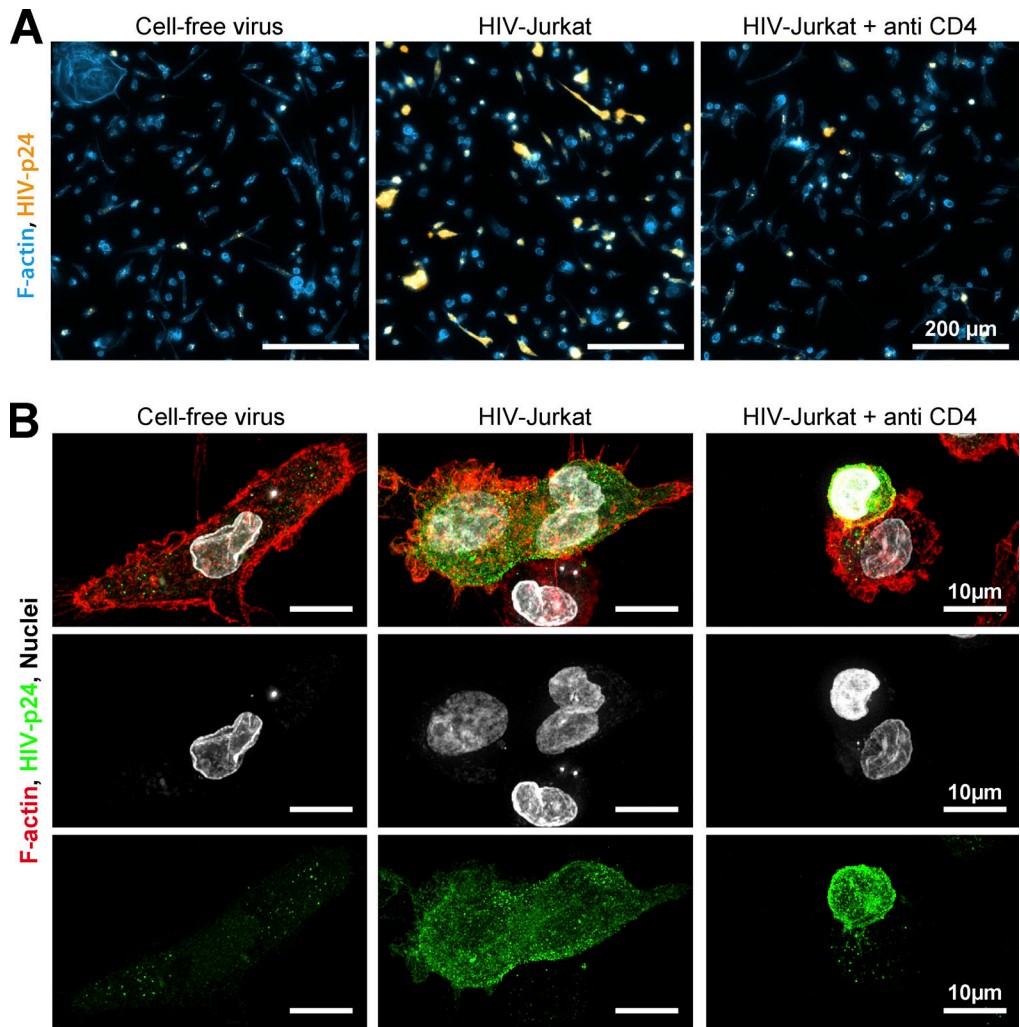

Figure S4.  **Related to** Fig. 2. **(A)** Ex vivo infection of purified tissue macrophages isolated from first trimester placenta and co-cultured for 24 h with either cell-free virus, or infected Jurkat cells in presence or not of neutralizing anti-CD4 antibody and stained for HIV-p24 (yellow) and F-actin (phalloidin, blue). Scale bars, 200 µm. **(B)** Macrophages under the same conditions as in (A) stained for HIV-p24 (green), F-actin (phalloidin, red), and nuclei (DRAQ5, gray) and analyzed by confocal microscopy. Scale bars, 10 µm.

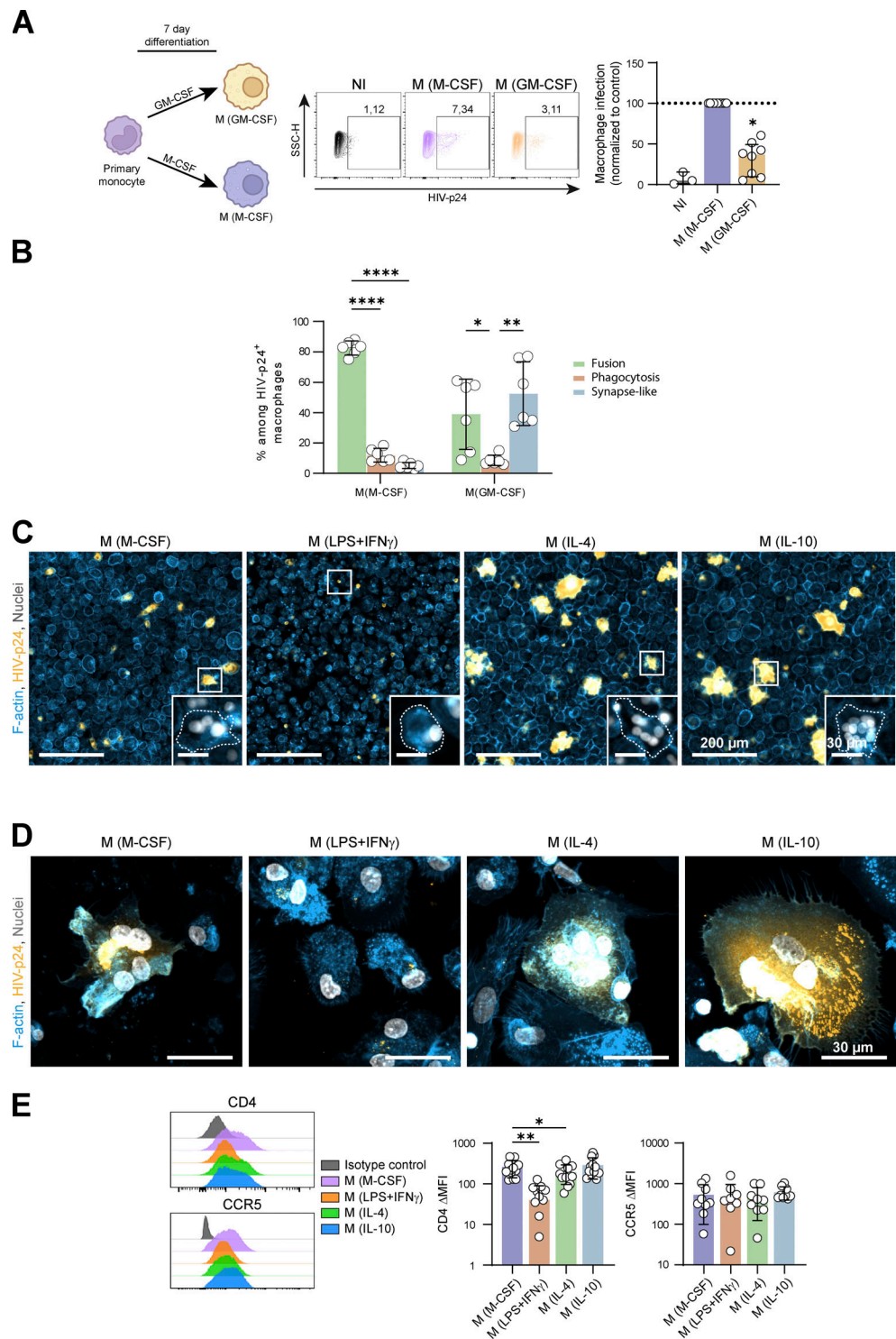

Figure S5.   **Related to** Fig. 3. **(A and B)** Analysis of infection of MDMs differentiated with M-CSF (M(M-CSF)) or GM-CSF (M(GM-CSF)) for 7 d, after a 24 h-co-culture with uninfected (NI) or infected Jurkat cells. **(A)** Left: Experimental design. Middle: Representative dot plots of HIV-1 p24 signals and gating strategy for selection of infected MDMs. Right: Quantification of the % of HIV-1-p24 + MDMs, normalized to M(M-CSF) condition (*n* = 8 donors, median ± interquartile range). **(B)** Quantification of the three mechanisms of HIV-1 transfer from infected Jurkat cells to polarized MDMs (as characterized in Fig.S1, C–E) among infected MDMs (*n* = 6 donors, mean ± SD). **(C)** Representative mosaic images of microscopy analysis of polarized (M(M-CSF), M(LPS+IFNγ), M(IL-4)) or M(IL-10) as in Fig. 3 A) MDM infection after a 24 h-co-culture with infected Jurkat cells. F-actin (blue), HIV-p24 (yellow), and nuclei (DAPI, gray). Inserts show magnifications of the white squares. Scale bars, 200 and 30 µm. **(D)** Representative microscopy images of polarized MDM infection after a 24 h-co-culture with infected autologous primary CD4[+] T cells. F-actin (blue), HIV-p24 (yellow) and nuclei (DAPI, gray). Scale bars, 30 µm. **(E)** Expression levels of CD4 and CCR5 at the surface of polarized MDMs by flow cytometry. Left: Representative profiles. Right: Quantification expressed as ΔMFI (*n* = 8 to 10 donors, mean ± SD). Statistical analyses: Multiple comparison test (A) Kruskal-Wallis and Dunn's, (B) two-way Anova and Tukey, and (E) one-way Anova and Tukey. * P ≤ 0.05; ** P ≤ 0.01; **** P ≤ 0.0001. Source data are available for this figure: SourceData FS5.

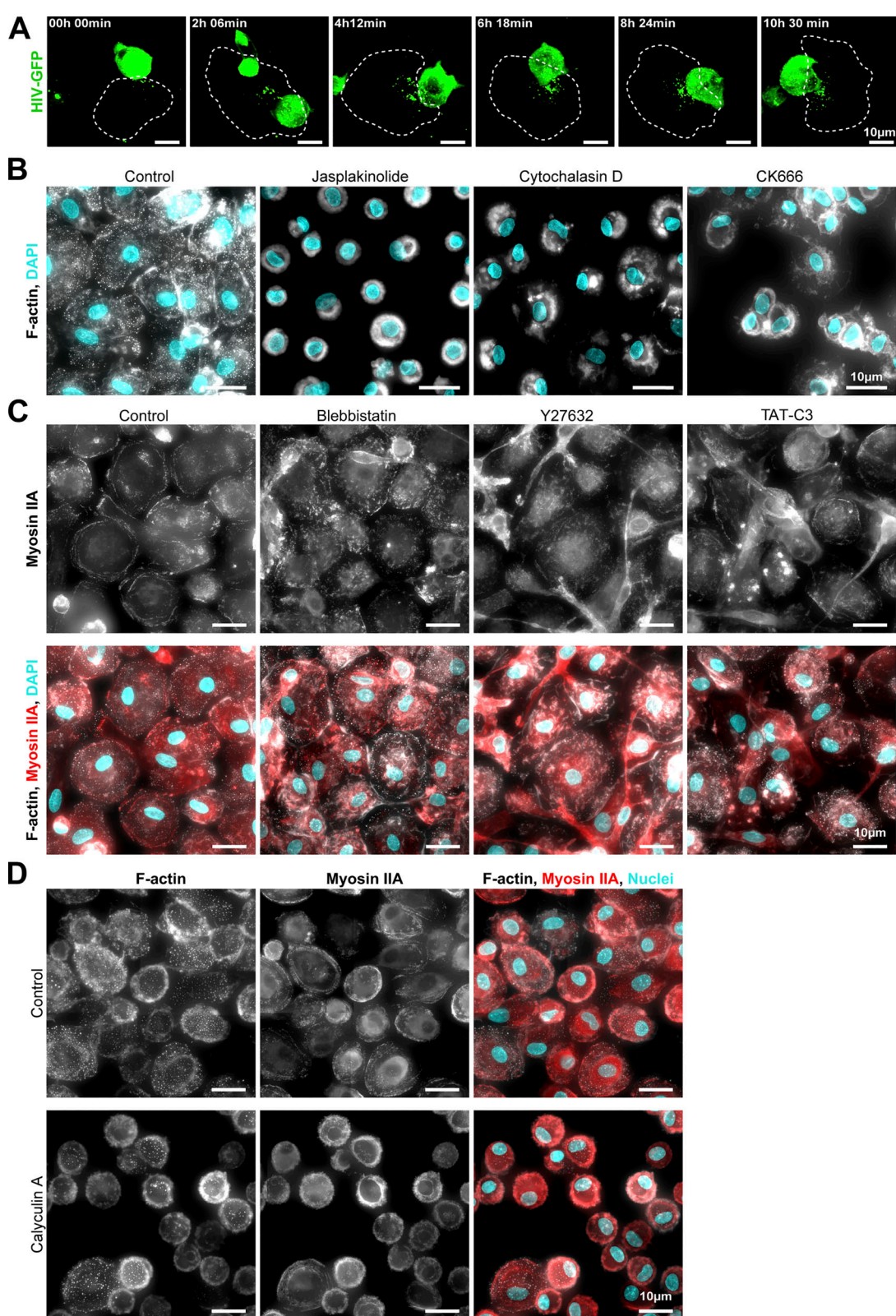

Figure S6.   **Related to** Fig. 4. **(A)** Time-lapse images of MDM infection by the synapse-like mechanism: MDMs were co-cultured with HIV-GFP-infected Jurkat cells as in Fig. 4 A (see also Video 5). HIV-GFP in green, dashed lines delineate the MDM. Scale bars, 10 µm. Representative images of more than 10 synapse-like contacts. **(B)** Representative images of F-actin cytoskeleton of MDMs treated different drugs (Jasplakinolide, Cytochalasin D, or CK666) or not (Control). F-actin (phalloidin, gray) and nuclei (DAPI, cyan). Scale bars, 10 µm. **(C)** Representative images of actomyosin cytoskeleton of MDMs treated different drugs (Blebbistatin, Y27632, or TAT-C3) or not (Control). F-actin (phalloidin, gray), Myosin IIA (red) and nuclei (DAPI, cyan). Scale bars, 10 µm. **(D)** Same analysis as in (C), except that MDMs were treated with Calyculin A or not (Control). Scale bars, 10 µm. **(C and D)** These images come from a mosaic (stitched together by the microscope).

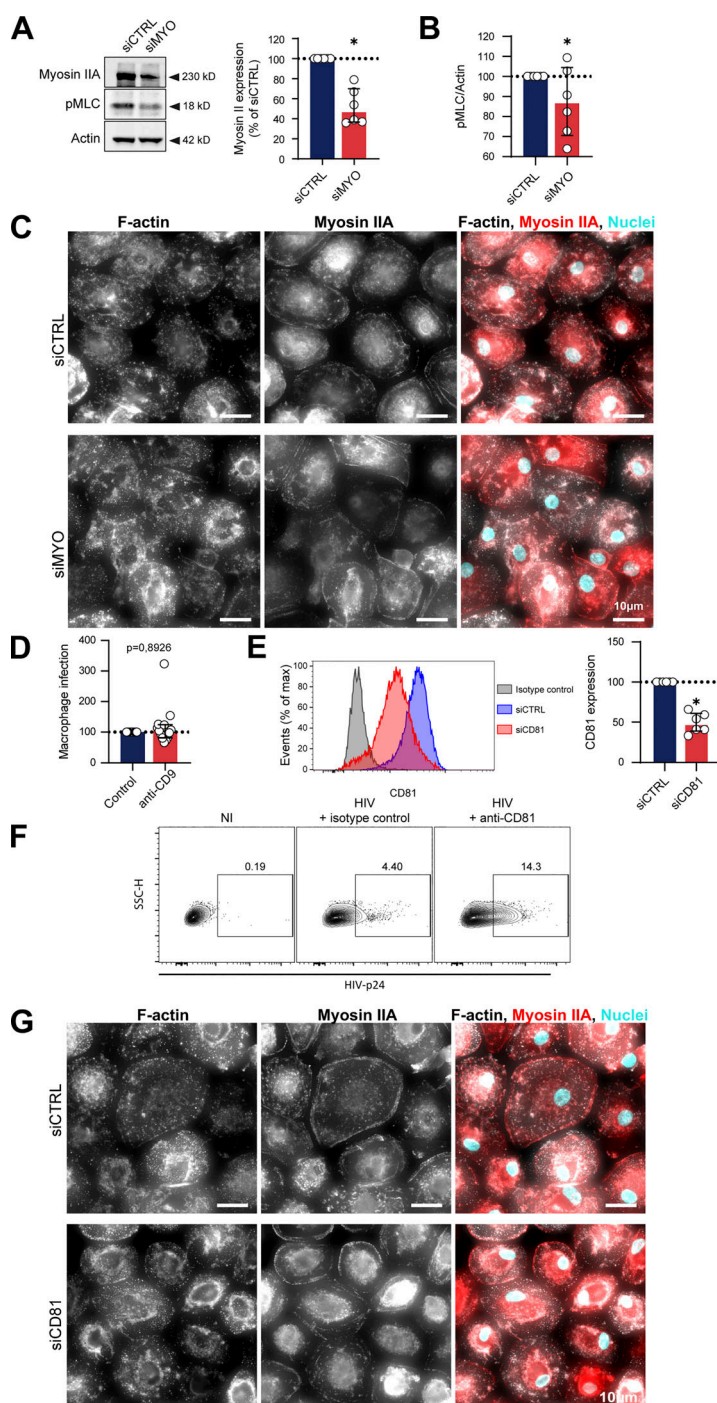

Figure S7. **Related to** Figs. 4 and 5**. (A–C)** Related to Fig. 4. **(A)** Analysis of Myosin expression and MLC phosphorylation by Western blot: MDMs were transfected with non-targeting siRNA (siCTRL) or targeting Myosin IIA (siMYO). Left: Representative images of Western blot analysis of the expression of Myosin IIA (top) and p-MLC (middle), with actin as loading control (bottom). Right: Quantification of Myosin IIA/actin, normalized to the siRNA control condition (*n* = 6 donors, median ± interquartile range). **(B)** Quantification of p-MLC/actin, normalized to the siCTRL condition (*n* = 6 donors, median ± interquartile range). **(C)** Representative images of actomyosin cytoskeleton of MDMs treated with siCTL or siMYO. F-actin (phalloidin, gray), Myosin IIA (red), and nuclei (DAPI, cyan). Scale bars, 10 μm. **(D–G)** Related to Fig. 5. **(D)** Quantification by flow cytometry of infection of MDMs incubated with isotype control (Control) or antibody targeting CD9 (anti-CD9) and co-cultured with infected Jurkat cells, normalized to the isotype control condition (*n* = 10 donors, median ± interquartile range). **(E)** Analysis of CD81 depletion by flow cytometry: MDMs were transfected with non-targeting siRNA (siCTRL) or targeting CD81 (siCD81). Left: Representative dot plot of CD81 signals. Right: Quantification of CD81 expression, normalized to the siCTRL condition (*n* = 6 donors, mean ± SD). **(F)** Flow cytometry analysis of alveolar macrophages infection after a 24 h-co-culture with uninfected Jurkat cells (NI) or Jurkat cells infected for 2 d in presence of anti-CD81 antibody or corresponding isotype control. Representative dot plots of HIV-p24 signals and gating strategy for selection of infected cells. See quantification in Fig. 5 D. **(G)** Representative images of actomyosin cytoskeleton of MDMs treated with siCTL or siCD81. F-actin (phalloidin, gray), Myosin IIA (red), and nuclei (DAPI, cyan). **(G)** This image comes from a mosaic (stitched together by the microscope). Scale bars, 10 μm. Statistical analyses: (A, B, and D) Wilcoxon test and (E) Paired *t* test. * P ≤ 0.05. Source data are available for this figure: SourceData FS7.

Video 1.   **Related to** Fig. 1 B. 3D reconstitution of confocal microscopy images, showing a MDM (CellTracker violet, blue) positive for HIV-1-p24 (red) with a nuclei from Jurkat T cell (CellTracker green) after co-culture with HIV-1-infected Jurkat T cells. Multiple profile views are shown.

Video 2.   **Related to** Fig. 2 B. 3D reconstitution of confocal microscopy images, showing F-actin (phalloidin, blue), HIV-p24 (red) and nuclei from T cell (CellTracker Blue, green) of human alveolar macrophages after co-culture with HIV-1-infected Jurkat T cells. Multiple profile views are shown.

Video 3.   **Related to** Fig. 2 C. 3D reconstitution of confocal microscopy images, showing F-actin (phalloidin, blue), HIV-p24 (red) and nuclei from T cell (CellTracker Blue, green) of human macrophages isolated from tonsils after co-culture with HIV-1-infected Jurkat T cells. Multiple profile views are shown.

Video 4.   **Related to** Fig. 4 A. Time-lapse of confocal microscopy images showing the fusion of HIV-1-infected Jurkat T cells (HIV-GFP viruses, green) with MDMs (DIC). 1 image every 7 min.

Video 5   **Related to** Fig. S6 A. Time-lapse of confocal microscopy images showing synaptic contact between HIV-1-infected Jurkat T cells (HIV-GFP viruses, green) and MDMs (DIC). 1 image every 7 min.

