## [Peer Review File · The Journal of Cell Biology]

Productive HIV-1 infection of tissue macrophages by fusion with infected CD4+ T cells

Rémi Mascarau, Marie Woottum, Léa Fromont, Rémi Gence, Vincent Cantaloube-Ferrieu, Vahlas Zoï, Kevin Lévêque, Florent Bertrand, Thomas Beunon, Arnaud Métais, Hicham El Costa, Nabila Jabrane-Ferrat, Yohan Gallois, Nicolas Guibert, Jean-Luc Davignon, Gilles Favre, Isabelle Maridonneau-Parini, Renaud Poincloux, Bernard Lagane, Serge Bénichou, Brigitte Raynaud-Messina, and Christel Verollet

Corresponding Author(s): Christel Verollet, Institut of Pharmacology and Structural Biology and Brigitte Raynaud-Messina, IPBS

Review Timeline:

Submission Date:	2022-05-20
Editorial Decision:	2022-06-27
Revision Received:	2022-12-05
Editorial Decision:	2023-01-19
Revision Received:	2023-01-31

Monitoring Editor: Ana-María Lennon-Dumenil

Scientific Editor: Lucia Morgado-Palacin

Transaction Report:

DOI: <https://doi.org/10.1083/jcb.202205103>

June 27, 2022

Re: JCB manuscript #202205103

Dr. Christel Verollet
Institut of Pharmacology and Structural Biology
CNRS UM5089 205 route de Narbonne
Toulouse 31077
France

Dear Dr. Verollet,

Thank you for submitting your manuscript entitled "Productive HIV-1 infection of tissue macrophages by fusion with infected CD4+ T cells". Your manuscript has been assessed by expert reviewers, whose comments are appended below. Although the reviewers express potential interest in this work, significant concerns unfortunately preclude publication of the current version of the manuscript in JCB.

You will see that the reviewers raise three major overlapping concerns: lack of proper controls and quantifications, inadequate data supporting cell-cell fusion and insufficient depth of the mechanism of HIV-induced cell-cell fusion.

In particular, there are missing controls for the fusion events (negative controls), for the siRNAs efficiency and for the specificity of blebbistatin treatment and myosin IIA depletion (rev #1 paragraphs 3 and 7 & rev #3 points 3-4), as well as no quantification of the different infection mechanisms (rev #1 paragraph 6 & rev #3 point 2) -all these technical issues need to be thoroughly addressed. Further, the extension of macrophage infection by cell fusion to other HIV tropic strains seems unclear as only the R5 strain is used, as noted by reviewer #2 in their point 3, thus it would be necessary that you repeat key experiments in another HIV tropic strain, such as X4, as suggested by this reviewer.

We agree with reviewer #1 (paragraph 2) that the methodology used to show cell-cell fusion seems inadequate as confocal microscopy does not allow for the accurate discrimination between juxtaposition of cells and cell-cell fusion -this reviewer proposes to adapt the BLAM enzymatic assay to your system to monitor and quantify cell-cell fusion more accurately by flow cytometry.

In our view, the reviewer requests on the mechanism of HIV-induced cell-cell fusion are reasonable and thus we expect that you address them in full. Reviewer #1 (paragraph 6) requests that you assess whether the polarization of macrophages indeed affects cell-cell fusion, as only infection is measured, and that you further develop the implication of the RhoA/ROCK/actomyosin pathway (paragraph 7). Reviewer #2 (point 2) would like that you assess the contribution of viral glycoprotein expression on the infected T cells to heterotypic fusion.

In addition, reviewers #2 and #3 find unclear the in vivo relevance of the findings. Reviewer #2 (point 1) is unclear to which extent the proposed mechanism contributes to HIV dissemination during infection in vivo and requests that you at least show that multinucleated giant cells of HIV-infected patients express T cell receptor. Along the same lines, reviewer #3 (point 1) questions the actual relevance of T cell viability and of the macrophage polarization state and asks to test whether supernatants from HIV-infected macrophages upon cell-cell fusion in these settings are indeed infectious by applying these supernatants to TZM-bl indicator cells. It would be required that you address these reviewer concerns to solidify the in vivo relevance of your work.

Despite some of your conclusions seem compromised in light of a recent published paper (Han et al., Plos Pathogens 2022), as noted by reviewer #1, we think that the study will still be of interest to our cell biology community.

Please let us know if you are able to address the major issues outlined above and wish to submit a revised manuscript to JCB. Note that a substantial amount of additional experimental data likely would be needed to satisfactorily address the concerns of the reviewers. The typical timeframe for revisions is three to four months. While most universities and institutes have reopened labs and allowed researchers to begin working at nearly pre-pandemic levels, we at JCB realize that the lingering effects of the COVID-19 pandemic may still be impacting some aspects of your work, including the acquisition of equipment and reagents. Therefore, if you anticipate any difficulties in meeting this aforementioned revision time limit, please contact us and we can work with you to find an appropriate time frame for resubmission. Please note that papers are generally considered through only one revision cycle, so any revised manuscript will likely be either accepted or rejected.

If you choose to revise and resubmit your manuscript, please also attend to the following editorial points. Please direct any editorial questions to the journal office.

GENERAL GUIDELINES:

Text limits: Character count is < 40,000, not including spaces. Count includes title page, abstract, introduction, results, discussion, and acknowledgments. Count does not include materials and methods, figure legends, references, tables, or supplemental legends.

Figures: Your manuscript may have up to 10 main text figures. To avoid delays in production, figures must be prepared according to the policies outlined in our Instructions to Authors, under Data Presentation, <https://jcb.rupress.org/site/misc/ifora.xhtml>. All figures in accepted manuscripts will be screened prior to publication.

IMPORTANT: It is JCB policy that if requested, original data images must be made available. Failure to provide original images upon request will result in unavoidable delays in publication. Please ensure that you have access to all original microscopy and blot data images before submitting your revision.

Supplemental information: There are strict limits on the allowable amount of supplemental data. Your manuscript may have up to 5 supplemental figures. Up to 10 supplemental videos or flash animations are allowed. A summary of all supplemental material should appear at the end of the Materials and methods section.

Please note that JCB now requires authors to submit Source Data used to generate figures containing gels and Western blots with all revised manuscripts. This Source Data consists of fully uncropped and unprocessed images for each gel/blot displayed in the main and supplemental figures. Since your paper includes cropped gel and/or blot images, please be sure to provide one Source Data file for each figure that contains gels and/or blots along with your revised manuscript files. File names for Source Data figures should be alphanumeric without any spaces or special characters (i.e., SourceDataF#, where F# refers to the associated main figure number or SourceDataFS# for those associated with Supplementary figures). The lanes of the gels/blots should be labeled as they are in the associated figure, the place where cropping was applied should be marked (with a box), and molecular weight/size standards should be labeled wherever possible. Source Data files will be made available to reviewers during evaluation of revised manuscripts and, if your paper is eventually published in JCB, the files will be directly linked to specific figures in the published article.

If you choose to resubmit, please include a cover letter addressing the reviewers' comments point by point. Please also highlight all changes in the text of the manuscript.

Regardless of how you choose to proceed, we hope that the comments below will prove constructive as your work progresses. We would be happy to discuss them further once you've had a chance to consider the points raised. You can contact the journal office with any questions, cellbio@rockefeller.edu.

Thank you for thinking of JCB as an appropriate place to publish your work.

Sincerely,

Ana-María Lennon-Dumenil
Monitoring Editor
Journal of Cell Biology

Lucia Morgado-Palacin, PhD
Scientific Editor
Journal of Cell Biology

Reviewer #1 (Comments to the Authors (Required)):

The study of Mascarau et al. focuses on the process involved in the infection of tissue macrophages by HIV-1 in vivo. They aim to tackle the following paradox: while HIV-1-infected macrophages are present in many tissues of infected patients where they may constitute viral reservoirs, macrophages remain relatively poorly permissive in vitro to cell-free HIV-1 infection. Cell-to-cell transfer of HIV is probably the most efficient way for the virus to infect the macrophages. Following previous recent studies including their own, the authors promote the idea that heterotypic cell fusion between infected CD4+ T cells and macrophages would be the most relevant in vivo with tissue resident macrophages. Using MDM (monocyte-derived macrophages) and tissue resident sorted macrophages cocultured with Jurkat T cells or CD4+ T cells infected in vitro, they analyzed the cell-cell HIV transfer processes, concluding that it mainly occurs through heterotypic cell fusion and tried to

decipher of the molecular mechanism involved.

The paper is well written and presented. The authors mainly used MDM and importantly, sorted tissue resident macrophages from various tissues, as well as MDM polarized into M1 or M2 phenotype. However, I have several important concerns about the ways the experiments were designed and the lack of controls. As compared to previous studies performed by the team and others, the findings presented appear limited. The authors tend to overinterpret their in vitro data to draw conclusions about what could happen in vivo. Overall, I do not find the study of sufficient novelty and quality to deserve to be published in the Journal of Cell Biology.

Main concerns

Cell-cell fusion. The study is mainly focused on fusion that could occur between HIV-1-infected Jurkat T cells and MDM under various conditions. However, the fusion per se is not measured but deduced from confocal microscopy analyses using differently labeled Jurkat (with Cell tracker) and MDM (with LysoTracker). It is very difficult to be sure, based on imaging, that cell-cell fusion rather than apposition for instance has occurred. Several of the images presented are indeed not convincing. See for instance, Fig2 depicting a macrophage supposed to have fused with CD3+ T cells. The staining of CD3 appears fuzzy in all the cytosol and even over the nuclei while it should be only associated with membranes. There is no quantification of the images presented in Fig 2.

Fusion of HIV-1 particles with target cells has been frequently measured using the so-called BLaM assay (Nat Biotech. 2002. 20(11):1151). An adaptation of this enzymatic assay using T cells infected with an HIV-1 BLaM exposed to MDM loaded with CF4 should allow to measure and quantify with accuracy by flow cytometry numerous events of cell-cell fusion.

Importantly, negative controls of fusion should have been added in the fusion assays performed by the authors, i.e. molecules (T20) or antibodies (anti-gp120, anti CD4, etc) known to inhibit fusion. Moreover, a comparison in the fusion assay performed by the authors without or with RT inhibitors (AZT Nevirapin) added at the time of mixing the two cell types would allow to clearly distinguish infection of MDM by viral particles released from or carried by infected Jurkat T cells from genuine fusion between both cell types leading to Gag staining present in the cytosol of the heterokaryons.

The use of Jurkat cells infected for 7 days which are pre-apoptotic does not represent an appropriate negative control. Use of Jurkat cells infected for shorter periods of time could have revealed if the capacity of fusion is transitory.

A recently published Plos Path paper (<https://doi.org/10.1371/journal.ppat.1010335>) contains very similar results to the ones presented by Mascarau et al. and somehow diminishes further the originality of the present study.

Testing the impact of polarization of macrophages on their susceptibility to infection, the authors do not show any link with fusion in contrast to their conclusion. Fig 3 panels C, D and E present quantifications achieved by p24+ cell analysis by FACS. In the results, authors stated that "compared to GM-CSF, M-CSF-driven conditions strongly favored HIV-1 infection of MDMs by heterotypic fusion with infected T cells (Fig. S3A)". Fig. S3A is a FACS dot-plot (p24 staining) cannot be used to draw a conclusion about the type of cell-cell transfer. The conclusion comes from the interpretation of Fig S3B/C/D. The figure S3 should contain a quantification of the different infection mechanisms characterized in S3B microscopy data, as in Fig.1. Importantly, the fusion index from Fig. S3D is not explained in the Material & Methods.

There is no control of the silencing efficiency for CD81 in Jurkat cells (Fig 5). The effect of the siRNA CD81 is hardly visible while the Ab anti CD81 exhibits a surprising positive and important effect on macrophage infection via infected Jurkat cells. This appears somehow contradictory. The implication of RhoA/ROCK/actomyosin pathway is not surprising but insufficiently documented experimentally. Overall, the mechanism underlying heterotypic cell fusion remains obscure.

Minor concerns:

The authors observed cell-cell fusion events between Jurkat cells and tissue-resident macrophages but did not extend this finding to autologous CD4+T lymphocytes (Fig 2). Similarly, it is unclear whether the authors confirmed their findings using MDM with autologous primary CD4+ T lymphocytes (Figure 1).

In the introduction the authors quote papers a little bit abusively to support a preeminent role of macrophages during infection in human.

The MDM appear seeded at a relatively high density from the images provided which can favor natural homotypic fusion between macrophages and may affect the results.

Sometimes in the figures, their titles, or in the legends CD4 T cells are mentioned while the authors actually used Jurkat cells.

Reviewer #2 (Comments to the Authors (Required)):

The authors demonstrate that T cells infected with a R5 tropic strain of HIV fuse with different macrophage populations, including tissue resident macrophages. Infection by this mechanism is dependent on the actin cytoskeleton and moderate myosin activity. The tetraspanin CD81 can inhibit HIV infection by fusion through its regulation of RhoA-ROCK/Myosin II. The authors argue that such fusion events play an important role for HIV transmission to long-lived macrophage populations during infection.

The authors describe an interesting mechanism of HIV transmission from infected T cells to macrophage populations. It remains, however, unclear to which extent this mechanism contributes to HIV dissemination during infection in vivo.

Major comments:

1. Due to previous studies that reported evidence for myeloid fusion in patients, resulting in multinucleated giant cells, any evidence for T cell marker expression on these could significantly strengthen the in vivo relevance of the described mechanism. Do multinucleated giant cells (MGCs) in patients' tissues display any signs of CD4+ T cell fusion, such as T cell receptor acquisition? Such in vivo evidence seems particularly important because the authors suggest that T cell fusion with macrophages is most efficient under non-inflammatory conditions that might not attract many T cells with and without HIV infection into tissues.
2. The authors have described in some detail the cytoskeletal requirements for the observed HIV infection by T cell fusion with macrophages. Are the observed fusion events between HIV infected T cells and tissue macrophages dependent on viral glycoprotein expression on the infected T cells? Can they be blocked by antibodies against CD4 and/or chemokine receptors? Due to the used HIV strain, it would be particularly interesting if CCR5 is required for the observed fusion.
3. In order to generalize the reported findings, it would also be important to document these with more than one HIV strain. Does the observed macrophage infection also occur with other HIV strains, in particular X4 tropic strains?

Reviewer #3 (Comments to the Authors (Required)):

In this manuscript Mascarau et al. have investigated the mechanisms by which HIV-1 transmits from infected CD4+ T cells to macrophages. They found that fusion of infected CD4+ T cell with human macrophages, including several tissue macrophage populations, is the main route for their productive infection. They further show that this mode of infection is modulated by the macrophage polarization state involving a specific short-lived adhesion structure controlled by CD81 tetraspanin via the RhoA-ROCK/Myosin II axis.

While it has long been known that cell-to-cell transmission of HIV-1 is the determinant mode of infection in tissue macrophages, the mechanisms involved in macrophage infection remains poorly understood. This in combination with the importance of tissue macrophage as major HIV-1 reservoirs makes this study timely and of high interest to the field. The majority of the data is of high quality and the usage of primary cells and tissue macrophages is a big plus as most of HIV studies are based on findings in immortalized/biologically less relevant cell lines.

However, some of the experiments would benefit from some additional controls to solidify these interesting findings.

- In Fig. 1D and 3E, quantification of p24 concentration in the supernatants from HIV-infected MDM are used as the readout for productive infection. While this shows production of viral particles, it does not confirm that these particles are indeed infectious. These findings would be strengthened by applying these supernatants to TZM-bl indicator cells to validate that T cell viability (Fig. 1) and macrophage polarization (Fig. 3) play a role in production of extracellular infectious virus particles upon cell fusion.
- Fig. 2A-C and S2A show the relevance of HIV-induced cell fusion in several tissue macrophages by immunostaining and confocal microscopy. However, the authors only show one cell per panel. Multiple example images and/or quantifications are needed to validate the significance and reproducibility of these findings.
- Given that treatment with the broad phosphatase inhibitor Calyculin A inhibits the percentage of multinucleated HIV-p24+ cells, its effects on actin cytoskeleton/viability should be included in S4C and Fig. 4G as a control for the specificity of myosin II inhibitor Blebbistatin treatment, which results in alteration of actomyosin cytoskeleton and an increase in cell fusion.
- The authors suggest an inhibitory role of Myosin IIA in MDM fusion with infected T cells based upon depletion of Myosin IIA in these cells. These motor proteins play a critical role in many cellular processes such as intracellular trafficking, migration, cytokinesis etc. As such, data validating the integrity of cell shape and motility should be included to exclude an inhibitory role of Myosin IIA due to effects on other processes that might indirectly affect cell fusion.

Toulouse, December 5th, 2022

Subject: Revised manuscript #202205103

Dear Ana-María Lennon-Dumenil,
Dear Lucia Morgado-Palacin, and editorial staff,

On behalf of all authors, we would like to thank you for your consideration of our manuscript (#202205103) entitled “Productive HIV-1 infection of tissue macrophages by fusion with infected CD4+ T cells”, and for providing us the detailed reviewers’ comments and the opportunity to submit a revised version of our manuscript.

You will find below a point-by-point response to the comments of the editor and of the reviewers.

We consider that the revised version of our manuscript is greatly improved thanks to the comments of the reviewers. Thus, we hope for a positive outcome from this revision process.

Sincerely yours,

Christel Vérollet and Brigitte Raynaud-Messina

Institute of Pharmacology & Structural Biology, CNRS – Université de Toulouse UMR 5089
205 Route de Narbonne, F31077 Toulouse, France.
Tel. +33(0)5 61 17 54 56 and (0)5 61 17 59 10
verollet@ipbs.fr, raynaud@ipbs.fr

We thank the reviewers for their valuable comments, and the editor for giving us the opportunity to revise the manuscript. We address all the editor's and reviewer's comments in the detailed point-by-point response provided below.

In short, we raised the three major overlapping concerns made by the 3 reviewers by (i) performing new experiments with controls and quantifications, (ii) better validating the cell-cell fusion process and (iii) going deeper into the mechanisms of cell-cell fusion and the link between CD81 and actomyosin.

We prepared a revised version of the manuscript with changes highlighted in yellow. As you will notice, most of the Figures and Supplemental Figures (with 2 totally novel figures and many new panels) have been modified according to the comments of the reviewers; and we also provide additional Figures to the concern of the reviewers only, see Fig. R1-8. Of note, a new contributing author (Zoi Vahlas) was added for her help to strengthen the cell fusion process (new Fig. 1B), following reviewers' comments. We also change the place for two authors, Vincent Cantaloube-Ferrieu and Bernard Lagane, as they contribute strongly to the revision process, in particular by performing the Blam-vpr assay to answer to reviewer 1 (Fig. R3).

Editor's comments:

In particular, there are missing controls for the fusion events (negative controls), for the siRNAs efficiency and for the specificity of blebbistatin treatment and myosin IIA depletion (rev #1 paragraphs 3 and 7 & rev #3 points 3-4), as well as no quantification of the different infection mechanisms (rev #1 paragraph 6 & rev #3 point 2) -all these technical issues need to be thoroughly addressed. Further, the extension of macrophage infection by cell fusion to other HIV tropic strains seems unclear as only the R5 strain is used, as noted by reviewer #2 in their point 3, thus it would be necessary that you repeat key experiments in another HIV tropic strain, such as X4, as suggested by this reviewer.

As we mention to the reviewers below, all the requested controls have already been published by our colleague and co-author of the present manuscript, Dr. S. Benichou (see Bracq et al., J. Virol. 2017; Xie et al., mBio. 2017; Han M, et al, PLoS Pathog. 2022). We now better explain this in the introduction (see page 4, line 26-35) and discussion (see page 14, line 5-7) sections. In addition, some of these controls have been done for the new experiments performed during the revision process and are provided as figures for the reviewers (Fig. R1 and R4).

We agree with reviewer #1 (paragraph 2) that the methodology used to show cell-cell fusion seems inadequate as confocal microscopy does not allow for the accurate discrimination between juxtaposition of cells and cell-cell fusion -this reviewer proposes to adapt the BLaM enzymatic assay to your system to monitor and quantify cell-cell fusion more accurately by flow cytometry. In our view, the reviewer requests on the mechanism of HIV-induced cell-cell fusion are reasonable and thus we expect that you address them in full. Reviewer #1 (paragraph 6) requests that you assess whether the polarization of macrophages indeed affects cell-cell fusion, as only infection is measured, and that you further develop the implication of the RhoA/ROCK/actomyosin pathway (paragraph 7).

In the answer to reviewer 1, we now explain the different approaches that we use to quantify cell-to-cell fusion in all our papers on macrophage fusion (see Verollet et al., J Immunol, 2010; Verollet et al., Blood, 2015; Raynaud-Messina et al., PNAS, 2018; Souriant et al., 2019; Dupont et al., eLife, 2020). We also performed new experiments to further support the fusion process, in particular by using different cytoplasmic markers to quantify cell-to-cell fusion and by adapting the Blam-Vpr system (Fig. 1B and Fig. R1, R3).

We appreciate the reviewer's comment on the polarization. We made some clarifications in the text and the quantification of cell fusion (it was previously Fig. 3SA) was added in the main Fig. 3 in the new version of the manuscript.

Regarding the role of the RhoA/ROCK/actomyosin pathway, we totally agree. We better develop our results in the new version of the manuscript (see page 10) and we added new panels of results in the new Fig. S10, as explained to reviewer 1 below.

Reviewer #2 (point 2) would like that you assess the contribution of viral glycoprotein expression on the infected T cells to heterotypic fusion.

This point has already been investigated in a previous publication (see Bracq et al., J. Virol. 2017), and is now better described in the new version of the introduction. Moreover, we also used anti-CD4 blocking antibodies as a control in some of the key experiments of this study and during the revision (see Fig. S4B, S5A-B, Fig. R1 and R4).

In addition, reviewers #2 and #3 find unclear the in vivo relevance of the findings. Reviewer #2 (point 1) is unclear to which extent the proposed mechanism contributes to HIV dissemination during infection in vivo and requests that you at least show that multinucleated giant cells of HIV-infected patients express T cell receptor.

We fully agree that the in vivo relevance of this mechanism of macrophage infection by fusion with infected T cells is a very important question. However, both difficult access to samples from infected patients without treatment and the low frequency of multinucleated infected macrophages make this analysis a huge challenge. We performed experiments in order to sustain the relevance of the cell fusion process that we discuss as an answer to reviewer 2. We hope that you will understand the difficulty of such experiments.

Along the same lines, reviewer #3 (point 1) questions the actual relevance of T cell viability and of the macrophage polarization state and asks to test whether supernatants from HIV-infected macrophages upon cell-cell fusion in these settings are indeed infectious by applying these supernatants to TZM-bl indicator cells. It would be required that you address these reviewer concerns to solidify the in vivo relevance of your work.

We performed new experiments during the revision process to show that viruses produced by macrophages upon cell fusion are indeed infectious on TZM-bl cell lines (see answer to reviewer 3).

Despite some of your conclusions seem compromised in light of a recent published paper (Han et al., Plos Pathogens 2022), as noted by reviewer #1, we think that the study will still be of interest to our cell biology community.

We now better discuss this paper in the main text of the revised manuscript in the introduction (see page 4, line 26-35) and discussion (see page 14, line 5-7) sections, as well as the answer to reviewer 1.

Reviewer #1:

Main concerns

Cell-cell fusion. *The study is mainly focused on fusion that could occur between HIV-1-infected Jurkat T cells and MDM under various conditions. However, the fusion per se is not measured but deduced from confocal microscopy analyses using differently labeled Jurkat (with Cell tracker) and MDM (with LysoTracker). It is very difficult to be sure, based on imaging, that cell-cell fusion rather than apposition for instance has occurred. Several of the images presented are indeed not convincing. Fusion of HIV-1 particles with target cells has been frequently measured using the so-called BLaM assay (Nat Biotech. 2002. 20(11):1151). An adaptation of this enzymatic assay using T cells infected with an HIV-1 BLaM exposed to MDM loaded with CF4 should allow to measure and quantify with accuracy by flow cytometry numerous events of cell-cell fusion.*

We apologize if the cell fusion was not clear enough in the manuscript and figures of the previous version. Nonetheless, we have extensively studied the process of HIV-induced cell fusion and the formation of MGC (multinucleated giant cells) in the past years (see Verollet et al., J Immunol, 2010; Verollet et al., Blood, 2015; Raynaud-Messina et al., PNAS, 2018; Souriant et al., Cell Reports, 2019; Dupont et al., eLife, 2020). Furthermore, this heterotypic fusion between infected CD4 T cells and macrophages was demonstrated for the first time and clearly characterized by our colleague and co-author of this article: Dr. S. Benichou (see Bracq et al., J. Virol. 2017; Xie et al., mBio. 2017; Han M, et al, PLoS Pathog. 2022).

To be more precise, in the present manuscript, we use several approaches (see below) certifying that we are indeed studying a cell fusion process. In addition, we also completed them by additional experiments during the revision process.

IF analysis: We are used to observe and quantify cell fusion by immunofluorescence (IF) analysis thanks to (i) the homogeneous diffusion of HIV-p24 protein in the uninfected target cell and (ii) the presence of multiple nuclei in the cytoplasm of a single HIV positive cell (using confocal imaging). In addition, fusion parameters were measured on large images acquired by confocal microscopy of cells co-stained with DAPI and phalloidin. Each condition was quantified on a gallery of images representing 1000 to 3000 cells, as in the references provided above. In addition, in this study, the pre-loading of the T cells with CellTracker blue allowed to follow the fate of the T cell nucleus and to show, that after co-culture, the T nucleus was in the MGC formed (see Fig. 2B-C and Fig. S2A and C). In the new version, to provide further evidence that the T cell nucleus was indeed in the infected macrophage, we displayed the different positions of the Z-stacks of a representative image (see new Fig. 1B and associated movie 1). It was also the case for the infection of alveolar and tonsil macrophages, see Fig 2 B-C and new movies 2 and 3.

In addition, we now provide a zoom as an insert with DAPI staining in all the panels where IF analysis was shown (See Fig. 1H, 3G, 4D and F and Fig. S6C-D). We also performed new coculture experiments by pre-labeling Jurkat cells and MDMs with cytoplasmic CellTrackers of different colors (CellTracker green and CellTracker violet (shown in red) respectively) (Fig. 1B and Fig. R1A). Again, these experiments prove unambiguously that the cytoplasm of the two cells are mixed during cell fusion. This new experiment has been included as new Fig. 1B

in the revised version. We also quantified the number of nuclei for the different modes of HIV transfer to macrophages (see new Fig. S2D).

Video-microscopy on live cells: The movies we obtained (Fig. 4A and movie 4) with Jurkat cells infected with a GFP-expressing virus strain (ADA-GFP-VSVG) and sorted showed the rapid disappearance of the infected T cell concomitant with a massive and homogeneous GFP labelling of the cytoplasm of macrophages. This is consistent with heterotypic fusion of the two cells (see page 8, line 27).

FACS analysis: It was important to note that quantification of infection by FACS in our manuscript was always done by gating on the pool of highly infected macrophages. The majority (around 70%) of these highly infected (HIV-p24⁺) macrophages (CD11b⁺) were also positive for CD3 (see Fig. R2A). We have added this figure in the new Fig. S2E, and provide all these details in the Material and Method section. Furthermore, after sorting, most of these highly infected cells were multinucleated (see below, Fig. R2B-C, legend next page). These results showed that the infected macrophages quantified by this method were predominantly multinucleated and contained material from Jurkat cells.

Figure R2: A. MDMs infected by transfer from infected Jurkat cells are mostly CD3-positive. Flow cytometry analysis of MDMs after co-culture with uninfected (NI) or 2-azy infected Jurkat cells (HIV). *Left:* Representative dot plots of HIV-p24 and CD3 stainings and gating strategy to determine the % of CD3⁺ among HIV-p24⁺ MDMs. NI: non infected MDMs. *Right:* quantification, each circle represents a single donor. **B-C. Characterization of sorted infected MDMs after a 24h-co-culture with Jurkat cells infected with a GFP-expressing viral strain.** (B) Representative dot plots of GFP signals and gating strategy for selection of two populations of infected MDMs (GFP^{low} and GFP^{high}). (C) After sorting, these two MDM populations were stained for F-actin (phalloidin, grey) and nuclei (DAPI, blue). HIV-GFP intensity is shown in yellow (left) or in color scale (arbitrary units, right). Scale bars, 100µm.

Blam-Vpr assay: Finally, thanks to the very interesting advice from the reviewer, we decided to adapt the Blam-Vpr system to quantify heterotypic cell fusion. Briefly, Jurkat cells were infected with a Blam-Vpr R5-tropic pseudotyped virus and cocultured with CCF2-labeled MDMs. 4 hours after coculture, cleavage of CCF2 in MDMs has been analyzed by flow cytometry (as in Souriant et al., Cell Reports, 2019). This experiment showed that at least 20% of MDMs was infected by fusion and this was totally abrogated in presence of T20 fusion inhibitor or Maraviroc (Fig. R3). We thank the reviewer for this great idea, and, while we estimate that these results are not essential in the present manuscript, we will certainly use this new way to quantify MDM infection by fusion in the future in the lab.

Figure R3: Quantification of MDM infection by fusion with infected Jurkat cells using Blam-vpr assay. Jurkat cells were infected through spinoculation (2000g, 4°C, 1h) with JR-FL, carrying the fusion protein Beta lactamase-vpr (BLAM-vpr). The cells were incubated 2h at 37°C, washed twice and then again incubated at 37°C overnight. After washing, they were cocultured with the MDMs ± T20 or ± Maraviroc (Mvc) for 4h at 37°C. T cells were removed and the MDMs were loaded with the BLAM substrate, CCF2 according to the kit instruction. *Left:* Representative dot plots of CD11 and CCF2 stainings and gating strategy for selection of MDMs with cleaved CCF2. *Right:* quantification. n=2 independent donors.

See for instance, Fig2 depicting a macrophage supposed to have fused with CD3+ T cells. The staining of CD3 appears fuzzy in all the cytosol and even over the nuclei while it should be only associated with membranes.

Regarding the specific comment of the reviewer on Fig. 2, since the CD3 staining was performed under permeabilized conditions and that, at least in several lymphoma T lymphocyte cell lines such as in Jurkat cells (Meddens et al., Front. Immunol., 2018; Evnouchidou et al., Nature Com., 2020), membranous and cytoplasmic CD3 stainings were observed, it was not surprising that CD3 staining appeared fuzzy throughout the cytosol. In addition, Fig. 2 showed immunohistochemistry (IHC) analysis of tissue macrophages which is quite challenging and does not allow to discriminate specific intracellular localizations. Of note, the same intracellular localization of CD3 was previously obtained for MDMs infected by fusion with infected Jurkat or primary T cells (see Fig. 6 in Bracq et al., J. Virol. 2017).

There is no quantification of the images presented in Fig 2.

We totally agree with this comment.

For Fig. 2A (tissue macrophages), macrophage infiltration is really low in non-inflammatory synovial tissues and quadruple labeling by IHC is challenging. That is why we only used this experiment to demonstrate that fusion of infected Jurkat cell can occur with macrophages in

their tissue environment. Quantification was not possible in this context and would not be informative.

It was also unfortunately not possible to perform quantitative experiments with purified myeloid cells isolated from children's healthy tonsils which represent less than 1% of the total cell population (Smith et al., JoVE, 2020). However, we had the opportunity to perform new experiments to address this point.

First, we performed new experiment with alveolar macrophages (AM) incubated with CD4 blocking antibodies. Although it was not possible to do a quantitative study because of the scarcity of material, we now illustrate several events of fusion between AM and infected Jurkat T cells, but also the inability of AM to fuse with infected Jurkat T cells after pre-incubation with a CD4 antibody in the new Fig. S4B. Second, we test the effect of CD4 blocking antibodies on the fusion of placental macrophages and provided the quantification page 7, line 15 of the revised manuscript. We obtained a 75% inhibition of infection of placental macrophages when CD4 was blocked using antibodies (number of nuclei \geq 270).

Importantly, negative controls of fusion should have been added in the fusion assays performed by the authors, i.e. molecules (T20) or antibodies (anti-gp120, anti CD4, etc) known to inhibit fusion. Moreover, a comparison in the fusion assay performed by the authors without or with RT inhibitors (AZT Nevirapin) added at the time of mixing the two cell types would allow to clearly distinguish infection of MDM by viral particles released from or carried by infected Jurkat T cells from genuine fusion between both cell types leading to Gag staining present in the cytosol of the heterokaryons.

We understand this comment. However, we chose not to provide these negative controls in this study as all these experiments have already been done in the previous papers published by our colleague Dr. S Bénichou who is also co-author of the present manuscript (see Bracq et al., J. Virol. 2017; Xie et al., mBio. 2017 ; Han M, et al, PLoS Pathog. 2022).

Please see in Bracq et al., J. Virol. 2017: for anti-CD4 (Leu3a): Fig. 1E, for anti-gp120 (PGT128, 10-1074, NIH 45-46, PG16): Fig. 1E, for Maraviroc and T20: Fig. 1F. These experiments show that all the HIV entry inhibitors block infection of MDMs by cell-to-cell fusion and thus demonstrate that this mechanism of macrophage infection by fusion is dependent on gp120-CD4/coreceptor interaction.

Please see for AZT treatment: Fig. 2G (Bracq et al., J. Virol. 2017). This experiment shows that the initial stage of macrophage infection (6h after co-culture) is not affected by this treatment and thus does not depend on the intracellular steps of viral reverse transcription.

As asked by the reviewer, the new experiments we performed during the revision process were done in the presence of T20 or Maraviroc or after pre-incubation of MDMs with an anti-CD4 antibody. You can see the results in Fig. R1 (by IF with cytoplasmic CellTrackers of different colors) and Fig. R4 (A, by FACs and B, by IF). As previously shown in Bracq et al., the frequency of MDMs infected by fusion was drastically reduced after all these treatments.

Figure R4: Effects of Maraviroc, T20 and blocking CD4 antibodies on MDM infection through fusion with 2 day-infected Jurkat cells by flow cytometry (A) and immunofluorescence, IF (B). **A. Left:** Representative dot plots of HIV-p24 staining. **Right:** quantification, each point represents a single donor. n=3 independent donors. **B.** Representative images of microscopy analysis. F-actin (blue), HIV-p24 (yellow) and nuclei (DAPI, grey). Scale bar, 200 μ m. Inserts show a 3-fold magnification.

The use of Jurkat cells infected for 7 days which are pre-apoptotic does not represent an appropriate negative control. Use of Jurkat cells infected for shorter periods of time could have revealed if the capacity of fusion is transitory.

As requested, we performed new experiments and Jurkat cells were infected for 2, 4 and 7 days before cocultures with MDMs. The ratio of the different modes of MDM infection was then quantified as in Fig. 1C. As predicted by the results shown in Fig. S1A, we observed a progressive decline in Jurkat viability (90%, 50% and 15% at day 2, 4 and 7) corresponding to an increase in the phagocytosis vs fusion ratio (Fig. R5).

Figure R5: Quantification of the three mechanisms (fusion, phagocytosis and synapse-like as characterized in Fig. S2A) of HIV-1 transfer from infected Jurkat cells to MDMs, among infected MDMs after a 24h-co-culture with Jurkat cells infected for 2 days (HIV-2d), 4 days (HIV-4d) or 7 days (HIV-7d). Each circle represents a single donor. n=4.

A recently published Plos Path paper (<https://doi.org/10.1371/journal.ppat.1010335>) contains very similar results to the ones presented by Mascarau et al. and somehow diminishes further the originality of the present study.

We understand this comment as the mechanism of macrophage infection by fusion with infected CD4 T cells is also the studied mechanism in this recent paper published by our collaborators and co-authors (Han et al, PLoS Pathog. 2022). The question addressed in this paper was to determine whether viruses expressing different envelope glycoproteins with CCR5 and/or CXCR4 usage are able to infect macrophages through this mechanism. Here, we demonstrated the relevance of this viral transmission route in human tissue macrophages and elucidate the molecular mechanisms involved (in the macrophage target cell side) in the regulation of this mode of macrophage infection. We could not discuss this paper in the first

submission as it had not yet been accepted. In the revised version, we now cited and discussed the PLoS pathogen paper in the introduction (see page 4, line 26-35) and discussion (see page 14, line 5-7) sections.

Testing the impact of polarization of macrophages on their susceptibility to infection, the authors do not show any link with fusion in contrast to their conclusion. Fig 3 panels C, D and E present quantifications achieved by p24+ cell analysis by FACS. In the results, authors stated that “compared to GM-CSF, M-CSF-driven conditions strongly favored HIV-1 infection of MDMs by heterotypic fusion with infected T cells (Fig. S3A)”. Fig. S3A is a FACS dot-plot (p24 staining) cannot be used to draw a conclusion about the type of cell-cell transfer. The conclusion comes from the interpretation of Fig S3B/C/D. The figure S3 should contain a quantification of the different infection mechanisms characterized in S3B microscopy data, as in Fig.1. Importantly, the fusion index from Fig. S3D is not explained in the Material & Methods.

We appreciate the reviewer’s comment and made some clarifications in the text. We also now provide the quantification of the infection and fusion index (that were in Supplemental Figures) in the main figures; see Fig. 3D-E. In addition, we performed new experiments of infection of MDMs in different activation states: pro-inflammatory (GM-CSF) and anti-inflammatory (M-CSF) and quantified the different modes of infection, as shown in new Fig. S6B. See also the result section page 7, line 31-35 of the manuscript.

We also apologize for the omission of the definition of the fusion index, which is now correctly defined in the Material and Method section.

There is no control of the silencing efficiency for CD81 in Jurkat cells (Fig 5).

In Fig. 5, we assume that the annotation of the figure was confusing and we have modified the figures accordingly (see cartoons in Fig. 5A-D). Since our goal was to determine the mechanism responsible for fusion in target macrophages, CD81 silencing was only performed in MDMs prior to the co-culture with infected Jurkat cells (Fig. 5B). Depletion efficiency in MDMs was quantified by flow cytometry (see new Fig. S10B).

The effect of the siRNA CD81 is hardly visible while the Ab anti CD81 exhibits a surprising positive and important effect on macrophage infection via infected Jurkat cells. This appears somehow contradictory.

To assess the role of CD81 tetraspanin in macrophages, we used two complementary methods: pre-incubation with specific blocking antibodies (Fig. 5A) or siRNAs directed against CD81 (Fig. 5B). Although we pre-incubated macrophages before the co-culture, we are aware that neutralization with specific antibodies could also have an effect on infected T cells. The siRNA technique only has an impact on macrophages, but with a moderate depletion efficiency (about 50%) (Fig. S10B). Indeed, macrophages derived from primary monocytes are more difficult to transfect than cell lines, and the silencing efficacy is generally modest. Although these 2 methods were hardly comparable in their inhibition capacity, they both induced, with a different amplitude, an increase in MDM infection whatever the donor cells (Jurkat cells, Fig. 5A-B or primary CD4 T cells, Fig. 5C) or the target cells (in vitro differentiated, Fig. 5A-C or purified tissue macrophages, Fig. 5D).

The implication of RhoA/ROCK/actomyosin pathway is not surprising but insufficiently documented experimentally. Overall, the mechanism underlying heterotypic cell fusion remains obscure.

We agree with this comment. In order to answer to the reviewer, we better developed the mechanism described in Fig. 5 in the result section (see page 10). To this end, we performed new experiments to assess the effect of CD81 depletion on the actomyosin cytoskeleton by IF and provide these results as new Fig. S10D. We showed that “CD81 depletion in MDMs induced a slight disruption of cortical myosin labeling and no impact on podosome organization” (page 10, line 17-20).

Minor concerns:

The authors observed cell-cell fusion events between Jurkat cells and tissue-resident macrophages but did not extend this finding to autologous CD4+T lymphocytes (Fig 2). Similarly, it is unclear whether the authors confirmed their findings using MDM with autologous primary CD4+ T lymphocytes (Figure 1).

Regarding Fig. 2, unfortunately, it is not possible to repeat the experiments with primary CD4 T cells as they need to be autologous. Indeed, blood samples from the patients from whom the lung macrophages are collected are not available.

Regarding Fig. 1, as requested by the reviewer, we performed new experiments to confirm the results obtained with Jurkat cells with autologous primary CD4 T cells infected (see new Fig. 1F and Fig. S3C). Indeed, we confirmed both by cytometry and IF analyses that MDM infection with viable T cells was more efficient compared to infection with apoptotic T cells (see page 6, line 9-11).

In the introduction the authors quote papers a little bit abusively to support a preeminent role of macrophages during infection in human.

We understand the reviewer’s comment and dampened our interpretation of some papers on macrophage infection in the introduction (see modifications page 3).

The MDM appear seeded at a relatively high density from the images provided which can favor natural homotypic fusion between macrophages and may affect the results.

Indeed, the density of MDMs used is quite high, but it allows a good MDM infection rate, as shown before in Verollet et al., J Immunol, 2010; Verollet et al., Blood, 2015; Souriant et al., Cell reports, 2019, Dupont et al., eLife, 2020, ... Under these conditions, MDMs are unable to perform homotypic fusion without addition of any cytokines or virus. For illustration, see representative IF images of MDMs (more than 1000 nuclei) in control conditions attesting the absence of fused macrophages (Fig. 1H-NI, Fig. S8 and S9C).

Sometimes in the figures, their titles, or in the legends CD4 T cells are mentioned while the authors actually used Jurkat cells.

We agree with this comment and we made the corrections in the entire manuscript and in the figures.

Reviewer #2:

Major comments:

1. Due to previous studies that reported evidence for myeloid fusion in patients, resulting in multinucleated giant cells, any evidence for T cell marker expression on these could significantly strengthen the in vivo relevance of the described mechanism. Do multinucleated giant cells (MGCs) in patients' tissues display any signs of CD4+ T cell fusion, such as T cell receptor acquisition? Such in vivo evidence seems particularly important because the authors suggest that T cell fusion with macrophages is most efficient under non-inflammatory conditions that might not attract many T cells with and without HIV infection into tissues.

We fully agree that the in vivo relevance of this mechanism of macrophage infection by fusion with infected T cells is a very important question. Before this paper, this mechanism has been only described in myeloid cells derived in vitro from human monocytes (Bracq et al., J. Virol. 2017; Xie et al., mBio. 2017 ; Raynaud-Messina B, et al., PNAS, 2018 ; Han M, et al., PLoS Pathog. 2022). That is why it was important for us to validate this fusion mechanism in tissue macrophages (Fig. 2, S4 and S5). We consider that it is already a big step to prove the relevance of this process.

To go further, although it is very difficult in Europe to obtain biopsies of HIV+ patients with a high viral load (i.e. without ART), we recently had the opportunity to obtain intestinal biopsies from an HIV-positive patient before and after a few days of ART treatment (from Dr. Delobel, department of Infectious and tropical diseases, Hôpital Purpan, Toulouse). In this tissue, we performed a quadruple IHC staining (Nuclei, HIV-p24, CD3 (CD4+ T cell marker) and CD68 (macrophage marker) as done in Fig. 2A. We obtained first encouraging results (see Fig. R6). However, these data remain preliminary; the major difficulties being the autofluorescence of the tissue (in the GFP channel) and the lack of sensitivity of p24 labelling in IHC. Considering that the human infected material is very precious and that not all infected macrophages contain detectable T cell material (deduced from study in SIV-infected macaques: Calantone et al., Immunity, 2014), it is now necessary to develop more accurate approach such as in situ hybridization with viral probes. This analysis unfortunately is not currently mastered in the team at the moment.

Figure R6: Immunohistological analyses of a human colon from an HIV+ patient: Nuclei (DAPI, blue), HIV-p24 (green), CD3 (CD4+ T cell marker, orange) and CD68 (macrophage marker, red). Scale bar, 200 μ m.

Besides consistent with this fusion mechanism, numerous studies have unambiguously shown the presence of multinucleated giant myeloid cells in all, if not almost all, tissues of the body (as example see Lewin-Smith M et al. Mod Pathol. 1999; Burdo TH et al. Immunol Rev. 2013; Dargent JL et al. Mod Pathol. 2000; Frankel SS et al. Science. 1996; Ryzhova EV et al. Virology. 2002; Teo I et al. J Virol. 1997; ...). In addition, two studies by the group of Brenchley have shown that, in non-human primates infected with SIV, myeloid cells contain viral DNA within tissues in which CD4+ T cells are abundant. Some of these infected myeloid cells contain T-cell material (measured by PCR quantification of rearranged TCR DNA) (Calantone et al., Immunity, 2014; DiNapoli et al., JCI Insight, 2017). The authors' interpretation is that the myeloid cells phagocytosed the infected T cells in vivo. Although this interpretation is consistent and properly supported, we believe that the cell fusion mechanism we describe in our study would also lead to the presence of T cell material in infected macrophages. To test this hypothesis during the revision, we performed a comparable PCR assay, using primers specific of TCR γ rearrangement in Jurkat cells (2 different pairs) and primers corresponding to non-rearranged TCR as control for MDM DNA. The qPCR analysis with control primers specific of non-rearranged DNA showed that the quality and quantity of MDM DNA were equivalent in the different experimental conditions. Importantly, as shown in Fig R7, we clearly detect DNA material from T cells in MDMs infected with 2 day-infected Jurkat cells (*i. e.* by fusion) compared to MDMs alone or in the presence of uninfected Jurkat cells. While indirectly addressing the reviewer question, these results allow to reinterpret Brenchley's results and propose that the in vivo observed infected myeloid cells that contain specific T cell material may correspond to phagocytosis events but also to fusion events with infected T cells. We now address more precisely this point in the discussion, see page 11, line 17 and page 13, line 27.

Figure R7: Total DNA was extracted from MDMs alone, MDMs cocultured with uninfected (NI) or infected (HIV+) Jurkat cells, non-infected and infected (HIV+) Jurkat cells, using the QIAamp DNA mini kit (Qiagen). qPCR was performed using the Quantitect SYBR green PCR Master mix (Roche) with 20 ng of DNA and 0.5 μ M primers (listed below). Amplification of DNAs was performed using the Applied Biosystems 7500 Real-Time PCR (Thermofisher). Each sample was amplified in triplicate. Relative expressions of TCR-rearranged (V8J1 and V11J1) and TCR-non rearranged were calculated by normalization to actin transcripts. Each point represents a single donor. n=2.

Séquences primers	TCR non-rearranged	Forward primer (TRGIP2)	TGAGAATCCCAGACCACCAGA
		Reverse primer (J1)	AGTGTGTCCACTGCCAAAGAG
	TCR rearranged (V8J1)	Forward primer (V8)	TTATGCAAGCACAGGGAAGAC
		Reverse primer (J1)	GTGTGTTCCTCCACTGCCAAAGAG
	TCR rearranged (V11J1)	Forward primer (V11)	GCTCAAGATTGCTCAGGTGGG
		Reverse primer (J1)	TGTTGTCCACTGCCAAAGAGTTT

2. The authors have described in some detail the cytoskeletal requirements for the observed HIV infection by T cell fusion with macrophages. Are the observed fusion events between HIV infected T cells and tissue macrophages dependent on viral glycoprotein expression on the infected T cells? Can they be blocked by antibodies against CD4 and/or chemokine receptors? Due to the used HIV strain, it would be particularly interesting if CCR5 is required for the observed fusion.

As mentioned in the answer to Reviewer 1, we agree that these control experiments are important, but we did not provide them again in this study as they have already been done in the previous papers of our colleague and co-author of this article: Dr. S. Benichou (see Bracq et al., J. Virol. 2017; Xie et al., mBio. 2017 ; Han M, et al, PLoS Pathog. 2022). Please see answer to reviewer 1 and additional experiments showing that, in our conditions, we obtained similar results (provided in Fig. R1 and R4). Concerning tissue macrophages, it was unfortunately not possible to perform quantitative experiments with purified myeloid cells isolated from children's healthy tonsils (representing less than 1% of the cell population) (Smith et al., JoVE, 2020). However, we performed new experiment with alveolar macrophages (AM) incubated with CD4 blocking antibodies. Although it is not possible to do a quantitative study because of the scarcity of material, we now illustrated the inability of AM to fuse with infected Jurkat T cells after pre-incubation with a CD4 antibody in the new Fig. S4B, lower panels (to compare with upper panels). Second, we quantified the effect of CD4 blocking antibodies on the fusion of placental macrophages and provided the results page 7, line 14-16 of the revised manuscript. We obtained a 75% inhibition of infection of placental macrophages when CD4 was blocked using antibodies (Nuclei number \geq 270). As decidual tissue was permissive to cell-free CCR5 HIV-1 strain but not infected by CXCR4 HIV-1 (Marlin et al., Plos one, 2009), we assume that this fusion is also CCR5 dependent.

3. In order to generalize the reported findings, it would also be important to document these with more than one HIV strain. Does the observed macrophage infection also occur with other HIV strains, in particular X4 tropic strains?

We thank the reviewer for raising this issue. In the first published paper, Bracq et al. (2017) showed that in contrast to CCR5-tropic strains, the X4 tropic strain NL4.3 was not efficiently transferred from T lymphocytes to MDMs by this cell-fusion mechanism. Then, in the recently published article of our collaborators and co-authors of the present study (Han et al., Plos Pathogens, 2022), it is nicely shown that this cell fusion process allows the infection of macrophages with R5 tropic strains that were otherwise considered as poorly macrophage-tropic in cell-free conditions. We better discussed these two important papers in the main text of the revised manuscript both in the introduction (page 4, line 26-35) and in the discussion (page 14, line 5-7).

Reviewer#3:

- In Fig. 1D and 3E, quantification of p24 concentration in the supernatants from HIV-infected MDM are used as the readout for productive infection. While this shows production of viral particles, it does not confirm that these particles are indeed infectious. These findings would be strengthened by applying these supernatants to TZM-bl indicator cells to validate that T cell viability (Fig. 1) and macrophage polarization (Fig. 3) play a role in production of extracellular infectious virus particles upon cell fusion.

We appreciate the comment of the reviewer. The requested experiments were performed by testing the ability of supernatants of MDMs to infect a TZM-bl indicator cell line and we compared virus infectivity at the same p24 concentration, as we have already done in Souriant et al., Cell Reports, 2019. Please see a representative IF analysis of TZM-bl infection in Fig. R8 below that was used to quantify infectivity (% of HIV-p24+ TZM-bl cells) and see details in the Method section. The results showed that, for supernatants corresponding to Fig. 1G: “The viruses produced by the fusion mechanism were infectious as they can infect the TZM-bl reporter cell line, and their infectivity was comparable to that of the few viruses produced by phagocytosis ($12.2 \pm 3.1\%$ of p24-positive TZM-bl vs. $9.3 \pm 4.5\%$, respectively; $n=3$)” and for supernatants corresponding to Fig. 3H : “The viruses produced in each condition were equally infectious on TZM-bl reporter cells”. See page 6, line 15 and page 8, line 13.

Figure R8: Representative IF images for the quantification of the infectivity of the released viral particles in the supernatant of MDMs co-cultured with 2 day-(A) or 7 day-(B) infected Jurkat cells. Supernatants of each condition (corresponding to experiments presented in Fig. 1G) were added on TZM-bl reporter cells, at equivalent concentration of p24, and rate of TZM-bl infection was quantified 48h latter by a semi-automatic quantification (nuclei number \geq 500, $n=3$ independent experiments). HIV-p24 (red) and nuclei (DAPI, green). Scale bar, 100 μ m

- Fig. 2A-C and S2A show the relevance of HIV-induced cell fusion in several tissue macrophages by immunostaining and confocal microscopy. However, the authors only show one cell per panel. Multiple example images and/or quantifications are needed to validate the significance and reproducibility of these findings.

We agree with this comment.

- For Fig. 2A and now Fig. S4A (Non-inflammatory synovial tissue): Macrophage infiltration is very low in these samples and quadruple labeling by IHC is challenging. That is why we only used this experiment to demonstrate that fusion of infected Jurkat cells can occur with macrophages in their tissue environment and provide only 2 images. Quantification was thus not possible in this context.

- Fig. 2B-C, new Fig. S4B and S5 (Purified tissue macrophages) : It was also unfortunately not possible to perform quantitative experiments with purified myeloid cells isolated from children's healthy tonsils which represent less than 1% of the total cell population (Smith et al., JoVE, 2020). However, we had the opportunity to perform new experiments with placental and alveolar macrophages to address this point. For placental macrophages, we have already provided a large image showing several fusion events (new Fig. S5). We also tested the effect of CD4 blocking antibodies on the fusion of these macrophages. In the new version, we provide the quantification page 7, line 14-16: Compared with infection of placental macrophages by transfer from viable Jurkat cells, "we observed a 75% inhibition in the infection index when macrophages were pre-treated with anti-CD4 antibodies (n>270 nuclei/condition)". For AMs, in the revised manuscript, we now show a gallery of images attesting that this mechanism of macrophage infection by fusion is not a rare event. We also performed new experiment by incubating AMs with CD4 blocking antibodies. Although it was not possible to do a quantitative study because of the scarcity of material and the fragility of these cells, we now illustrate several events of fusion between AMs and infected Jurkat T cells, as well as the inability of AMs to fuse with infected Jurkat T cells after pre-incubation with a CD4 antibody in the new Fig. S4B.

- Given that treatment with the broad phosphatase inhibitor Calyculin A inhibits the percentage of multinucleated HIV-p24+ cells, its effects on actin cytoskeleton/viability should be included in S4C and Fig. 4G as a control for the specificity of myosin II inhibitor Blebbistatin treatment, which results in alteration of actomyosin cytoskeleton and an increase in cell fusion.

For all the drugs used in this study, first experiments to determine the drug concentration that does not affect viability but has an impact on the actin cytoskeleton have been done systematically. On this basis, we chose 25µM for Blebbistatin and 5nM for Calyculin. We now provide the results in new Fig. S8C by showing IF images after Calyculin treatment, as already done for Blebbistatin treatment (new Fig. S8B-C). In agreement with previous works (Kolega J et al., MBC, 2006 and Labernadie et al., PNAS, 2010), they confirm that "upon Blebbistatin and Calyculin A treatments, MDMs showed an altered actomyosin cytoskeleton without any effect on cell viability or podosome formation" (page 9, line 15).

- The authors suggest an inhibitory role of Myosin IIA in MDM fusion with infected T cells based upon depletion of Myosin IIA in these cells. These motor proteins play a critical role in many cellular processes such as intracellular trafficking, migration, cytokinesis etc. As such, data validating the integrity of cell shape and motility should be included to exclude an inhibitory role of Myosin IIA due to effects on other processes that might indirectly affect cell fusion

As pointed out by the reviewer, we show myosin IIA has an inhibitory role in the fusion of MDMs with infected T cells, as upon depletion of myosin IIA expression, macrophage infection by fusion is increased. It is largely accepted that that this protein plays a critical role in many cellular processes including promoting cell motility (Heissler et al., Cell Mol Life Sci,

2013). Thus, we consider that even if inhibition of myosin IIA had an inhibitory effect on macrophage migration in our conditions, the observed effect on fusion would just be underestimated. Thus, the increase in fusion with infected T cells cannot be explained by the potential role of myosin on migration, and in all the cases, the effect of myosin IIA depletion on macrophage infection through heterotypic fusion appears dominant compared to the effect on migration. To answer to the reviewer on the other issues concerning cell shape after siRNA against myosin IIA, we performed additional experiments. These data are included in the new manuscript (New Fig. S9C). They clearly showed that “siRNA against myosin did not affect cell viability neither cell morphology (Fig. S9C)” (page 9, line 26).

January 19, 2023

RE: JCB Manuscript #202205103R

Dr. Christel Verollet
Institut of Pharmacology and Structural Biology
CNRS UM5089 205 route de Narbonne
Toulouse 31077
France

Dear Dr. Verollet:

Thank you for submitting your revised manuscript entitled "Productive HIV-1 infection of tissue macrophages by fusion with infected CD4+ T cells". We would be happy to publish your paper in JCB pending final revisions necessary to meet our formatting guidelines (see details below).

To avoid unnecessary delays in the acceptance and publication of your paper, please read the following information carefully. Please go through all the formatting points paying special attention to those marked with asterisks.

A. MANUSCRIPT ORGANIZATION AND FORMATTING:

Full guidelines are available on our Instructions for Authors page, <https://jcb.rupress.org/submission-guidelines#revised>.
Submission of a paper that does not conform to JCB guidelines will delay the acceptance of your manuscript.

1) Text limits: Character count for Articles and Tools is < 40,000, not including spaces. Count includes title page, abstract, introduction, results, discussion, and acknowledgments. Count does not include materials and methods, figure legends, references, tables, or supplemental legends.

2) Figures limits: Articles and Tools may have up to 10 main text figures.

Please note that main text figures should be provided as individual, editable files.

3) Figure formatting:

*** Molecular weight or nucleic acid size markers must be included on all gel electrophoresis. Please add MW markers to Figs 5E-F and S9A.

*** Scale bars must be present on all microscopy images, including inset magnifications. Please add scale bars to inset magnifications in Figs 1H, 3G, 4D, 4F and S6C.

Also, please avoid pairing red and green for images and graphs to ensure legibility for color-blind readers. If red and green are paired for images, please ensure that the particular red and green hues used in micrographs are distinctive with any of the colorblind types. If not, please modify colors accordingly or provide separate images of the individual channels.

4) Statistical analysis:

*** Error bars on graphic representations of numerical data must be clearly described in the figure legend.

*** The number of independent data points (n) represented in a graph must be indicated in the legend. Please, also indicate whether 'n' refers to technical or biological replicates (i.e. number of analyzed cells, samples or animals, number of independent experiments).

If independent experiments with multiple biological replicates have been performed, we recommend using distribution-reproducibility SuperPlots (please, see Lord et al., JCB 2020) to better display the distribution of the entire dataset, and report statistics (such as means, error bars, and P values) that address the reproducibility of the findings.

Statistical methods should be explained in full in the materials and methods in a separate section.

For figures presenting pooled data the statistical measure should be defined in the figure legends.

*** Please also be sure to indicate the statistical tests used in each of your experiments (both in the figure legend itself and in a separate methods section) as well as the parameters of the test (for example, if you ran a t-test, please indicate if it was one- or two-sided, etc.). Please, indicate the statistical tests used in the corresponding figure legends.

If you used parametric tests in your study (i.e. t-tests), you should have first determined whether the data was normally distributed before selecting that test. In the stats section of the methods, please indicate how you tested for normality. If you did not test for normality, you must state something to the effect that "Data distribution was assumed to be normal but this was not formally tested."

5) Abstract and title:

The abstract should be no longer than 160 words and should communicate the significance of the paper for a general audience.

The title should be less than 100 characters including spaces. Make the title concise but accessible to a general readership.

6) Materials and methods:

Should be comprehensive and not simply reference a previous publication for details on how an experiment was performed. The text should not refer to methods "...as previously described."

Also, the materials and methods should be included with the main manuscript text and not in the supplementary materials.

7) For all cell lines, vectors, constructs/cDNAs, etc. - all genetic material: please include database / vendor ID (e.g., Addgene, ATCC, etc.) or if unavailable, please briefly describe their basic genetic features, even if described in other published work or gifted to you by other investigators (and provide references where appropriate).

*** Please be sure to provide the sequences for all of your oligos: primers, si/shRNA, RNAi, gRNAs, etc. in the materials and methods. Please provide the sequences for your siRNA non-targeting control pool.

*** You must also indicate in the methods the source, species, and catalog numbers/vendor identifiers (where appropriate) for all of your antibodies, including secondary. If antibodies are not commercial, please add a reference citation if possible. Please, indicate the species and catalog numbers for all of your antibodies.

8) Microscope image acquisition:

*** The following information must be provided about the acquisition and processing of images:

- a. Make and model of microscope
- b. Type, magnification, and numerical aperture of the objective lenses
- c. Temperature
- d. imaging medium
- e. Fluorochromes
- f. Camera make and model
- g. Acquisition software
- h. Any software used for image processing subsequent to data acquisition. Please include details and types of operations involved (e.g., type of deconvolution, 3D reconstitutions, surface or volume rendering, gamma adjustments, etc.).

10) Supplemental materials:

*** There are strict limits on the allowable amount of supplemental data. Articles/Tools may have up to 5 supplemental figures. There is no limit for supplemental tables. Currently, you have 10 supplemental figures. We can give you a bit more space, but we would need you to try to reduce the number of supplementary figures (up to 6-7 if possible) by consolidating data from two figures into one or moving supplemental data to one of the main figures. Please be sure to correct the callouts in the text to reflect this change.

Please note that supplemental figures and tables should be provided as individual, editable files.

*** A summary of all supplemental material should appear at the end of the Materials and Methods section (please see any

recent JCB paper for an example of this summary).

11) Video legends:

*** Video legends should describe what is being shown, the cell type or tissue being viewed (including relevant cell treatments, concentration and duration, or transfection), the imaging method (e.g., time-lapse epifluorescence microscopy), what each color represents, how often frames were collected, the frames/second display rate, and the number of any figure that has related video stills or images.

12) eTOC summary:

A ~40-50 word summary that describes the context and significance of the findings for a general readership should be included on the title page.

*** The statement should be written in the present tense and refer to the work in the third person. It should begin with "First author name(s) et al..." to match our preferred style.

13) Conflict of interest statement:

JCB requires inclusion of a statement in the acknowledgements regarding competing financial interests. If no competing financial interests exist, please include the following statement: "The authors declare no competing financial interests."

14) A separate author contribution section is required following the Acknowledgments in all research manuscripts.

*** All authors should be mentioned and designated by their first and middle initials and full surnames and the CRediT nomenclature is encouraged (<https://casrai.org/credit/>).

15) ORCID IDs: ORCID IDs are unique identifiers allowing researchers to create a record of their various scholarly contributions in a single place. At resubmission of your final files, please consider providing an ORCID ID for as many contributing authors as possible.

16) Materials and data sharing:

All animal and human studies must be conducted in compliance with relevant local guidelines, such as the US Department of Health and Human Services Guide for the Care and Use of Laboratory Animals or MRC guidelines, and must be approved by the authors' Institutional Review Board(s). A statement to this effect with the name of the approving IRB(s) must be included in the Materials and Methods section.

*** As a condition of publication, authors must make protocols and unique materials (including, but not limited to, cloned DNAs; antibodies; bacterial, animal, or plant cells; and viruses) described in our published articles freely available upon request by researchers, who may use them in their own laboratory only. All materials must be made available on request and without undue delay. We strongly encourage to deposit all the cell lines/strains and reagents generated in this study in public repositories.

All datasets included in the manuscript must be available from the date of online publication, and the source code for all custom computational methods, apart from commercial software programs, must be made available either in a publicly available database or as supplemental materials hosted on the journal website. Numerous resources exist for data storage and sharing (see Data Deposition: <https://rupress.org/jcb/pages/data-deposition>), and you should choose the most appropriate venue based on your data type and/or community standard. If no appropriate specific database exists, please deposit your data to an appropriate publicly available database.

17) Please note that JCB now requires authors to submit Source Data used to generate figures containing gels and Western blots with all revised manuscripts. This Source Data consists of fully uncropped and unprocessed images for each gel/blot displayed in the main and supplemental figures. The Source Data files will be directly linked to specific figures in the published article.

Since your paper includes cropped gel and/or blot images, please be sure to provide one Source Data file for each figure that contains gels and/or blots along with your revised manuscript files. File names for Source Data figures should be alphanumeric without any spaces or special characters (i.e., SourceDataF#, where F# refers to the associated main figure number or SourceDataFS# for those associated with Supplementary figures). The lanes of the gels/blots should be labeled as they are in the associated figure, the place where cropping was applied should be marked (with a box), and molecular weight/size standards should be labeled wherever possible.

B. FINAL FILES:

Thank you for this interesting contribution, we look forward to publishing your paper in Journal of Cell Biology.

Sincerely,

Ana-Maria Lennon-Dumenil
Monitoring Editor
Journal of Cell Biology

Lucia Morgado-Palacin, PhD
Scientific Editor
Journal of Cell Biology

Reviewer #1 (Comments to the Authors (Required)):

The authors have addressed most of my concerns. Adding the BLAM assay and the various quantifications have strengthened the study.

Reviewer #2 (Comments to the Authors (Required)):

Manuscript Nr: 202205103R

Mascarau et al., "Productive HIV-1 infection of tissue macrophages by fusion with infected CD4+ T cells"

The authors demonstrate that T cells infected with a R5 tropic strain of HIV fuse with different macrophage populations, including tissue resident macrophages. Infection by this mechanism is dependent on the actin cytoskeleton and moderate myosin activity. The tetraspanin CD81 can inhibit HIV infection by fusion through its regulation of RhoA-ROCK/Myosin II. The authors argue that such fusion events play an important role for HIV transmission to long-lived macrophage populations during infection.

In their revised manuscript the authors have addressed most of my concerns by providing preliminary data that T cell markers might be detectable on multinucleated cells during HIV infection (Figure R6) and that rearranged TCR genes are transferred by T cell fusion with macrophages (Figure R7). Such TCR detection was previously reported for macrophages in HIV infected individuals and might indicate fusion events. Figure R7 could be incorporated into the supplemental information of the current manuscript. Furthermore, they provide evidence that CD4 specific antibodies block fusion and discuss their previous published findings that only R5 tropic strains are transferred by this mechanism. Therefore, I find the revised manuscript significantly improved.

Reviewer #3 (Comments to the Authors (Required)):

My concerns have been addressed satisfactory by both experiments and arguments.